# Effect of Intermittent Structures on the Spectral Index of Magnetic field in the Slow Solar Wind

Xin Wang[1,2], Xuanhao Fan[1], Yuxin Wang[1], Honghong Wu[3], and Lei Zhang[4]

[1]School of Space and Environment, Beihang University, Beijing, 100083, China
[2]Key Laboratory of Space Environment monitoring and Information Processing of MIIT
[3]School of Electronic Information, Wuhan University, Wuhan, 430072, China
[4]Qian Xuesen Laboratory of Space Technology, Beijing, 100094, China

**Correspondence:** Xin Wang (wangxinpku0209@gmail.com)

**Abstract.** Intermittent structures are ubiquitous in the solar wind turbulence, and they can significantly affect the power spectral index of magnetic field fluctuations which reflects the cascading process of the turbulence. However, an analytical relationship between intermittency level and the magnetic spectral index has not been shown yet. Here we present the continuous variation of the magnetic spectral index in the inertial range as a function of the intermittency level. By using the measurements from the WIND spacecraft, we find 42,272 intervals with different levels of intermittency and with duration of 5-6 minutes from 46 slow-wind streams between 2005 and 2013. Among them, each of the intermittent intervals is composed of one dominant intermittent structure and background turbulent fluctuations. For each interval, a magnetic spectral index $\alpha_B$ is determined for the Fourier spectrum of magnetic field fluctuations in the inertial range between 0.01 Hz and 0.3 Hz. A parameter $I_{max}$, which corresponds to the maximum of the trace of partial variance increments of the intermittent structure, is introduced as an indicator of the intermittency level. Our statistical result shows that as $I_{max}$ increases from 0 to 20, the magnetic spectrum becomes steeper gradually and the magnetic spectral index $\alpha_B$ decreases from $-1.63$ to $-2.01$. Accordingly, an empirical relation is established between $\alpha_B$ and $I_{max}$ for the first time as $\alpha_B = 0.4\exp(-I_{max}/5) - 2.02$. The result will help us to know more details about the contributions of the intermittent structures on the magnetic power spectra, and further about the physical nature of the energy cascade taking place in the solar wind. It will also help to improve the turbulence theories that contains intermittent structures.

## 1 Introduction

Intermittent structures are ubiquitous in the solar wind turbulence. They correspond to the long tail of the non-Gaussian probability distribution functions of plasma or field fluctuations (Burlaga, 1991b; Marsch and Tu, 1994, 1997). Previous studies have revealed that the intermittent structures are associated with current sheets and different types of discontinuities at small scales (tens of seconds) (Burlaga, 1969; Veltri and Mangeney, 1999; Servidio et al., 2012; Wang et al., 2013; Osman et al., 2014), and are associated with the boundary between two adjacent flux ropes at large scales (tens of minutes) (Bruno et al., 2001; Borovsky, 2008) in the solar wind turbulence. These structures play an important role in the turbulence cascading and dissipation processes (Tu and Marsch, 1995; Bruno and Carbone, 2013, and the references therein).

The intermittent structures with large-amplitude fluctuations make a substantial contribution to the shape and power level of magnetic-field spectra, which are directly related to the physical nature of the energy cascade taking place in the solar wind (Sari and Ness, 1969; Salem et al., 2007; Li et al., 2011; Borovsky, 2010). They often make the magnetic spectra become steeper (Siscoe et al., 1968; Burlaga, 1968; Salem et al., 2009). In previous studies, the time series of discontinuities were reported to produce $f^{-2}$ energy spectrum (Sari and Ness, 1969; Roberts and Goldstein, 1987; Champeney, 1973; Dallas and Alexakis, 2013) in the inertial range. Later, people also found that the discontinuities can also produce the magnetic power spectra that are shallower than $f^{-2}$. Li et al. (2011) studied the effect of current sheets on the magnetic power spectrum from the Ulysses Observations. They found that the current-sheet-abundance periods and current-sheet-free periods show $f^{-5/3}$ Kolmogorov scaling and $f^{-3/2}$ Iroshnikov-Kraichnan scaling, respectively. Accordingly, they proposed that the current sheet is the cause of the Kolmogorov scaling. This finding was confirmed by Borovsky (2010), who created an artificial time series that preserves the timing and amplitudes of the discontinuities from the ACE spacecraft observations. The artificial time series produces a magnetic power-law spectrum with a slope near the Kolmogorov $f^{-5/3}$ scaling in the inertial range. They emphasis that any interpretation of the dynamics or evolution of the solar wind turbulence should account for the contribution of strong discontinuities to the measurements. The intermittent structures can also lead to the anomalous (multifractal) scaling of structure functions (Veltri and Mangeney, 1999; Veltri, 1999; Salem et al., 2007, 2009).

The intermittent structures also influence the magnetic spectral anisotropy of the fluctuations in the solar wind turbulence. The magnetic spectral index of the magnetic field fluctuations was reported to be anisotropic with respect to the scale-dependent local mean field in the inertial range (Horbury et al., 2008; Podesta, 2009; Luo and Wu, 2010; Chen et al., 2011; Forman et al., 2011; Wicks et al., 2011). After removing the intermittency from the turbulence, Wang et al. (2014) found that the anisotropy of the magnetic spectral index turned out to nearly disappear. The magnetic spectrum in the parallel direction becomes shallower from $f^{-2}$ to $f^{-5/3}$, which is close to the scaling in the perpendicular direction. They concluded that the observed magnetic spectral anisotropy could result from intermittency. The result was confirmed by Telloni et al. (2019). Wang et al. (2015) made a comparison between the spectral anisotropy of magnetic fluctuations with low amplitude and that with moderate amplitude. The statistical results showed that the anisotropy is only present in the moderate-amplitude situation, and is absent for the low-amplitude cases. Accordingly, they suggested that the magnetic spectral anisotropy is dependent on the fluctuation amplitude. Later, Wu et al. (2020) presented an analysis on the scaling anisotropy with a stationary background field and found the same isotropy for the moderate-amplitude fluctuations after removing those intermittent structures. Through numerical simulation in three-dimensional Magnetohydrodynamic turbulence, Yang et al. (2017) found the influence of intermittency on the quasi-perpendicular scaling of magnetic field and velocity fluctuations.

Recently, the magnitude and thickness of the current sheets are also found to have significant effect on the power level in the dissipation range and the frequency location of the magnetic spectral break (Borovsky and Podesta, 2015; Borovsky and Burkholder, 2020; Podesta and Borovsky, 2016). People have also studied the heating effect of the intermittent structures through both observations (Osman et al., 2011, 2012b, a; Borovsky and Denton, 2011; Wang et al., 2013; Liu et al., 2019; Zhou et al., 2022) and simulations (Parashar et al., 2009; Servidio et al., 2012; Wan et al., 2012; Zhang et al., 2015).

From the previous studies mentioned above, people have realized that the intermittent structure is an important part of the solar wind turbulence, and it can significantly affect the shape and power level of the magnetic power spectrum. A close correspondence between intermittency and changes in the 2nd-order scaling properties has been well established. There is a huge literature on the correction of the scaling properties due to intermittency, and many improved cascade models have been proposed to revise the original Kolmogorov results. However, no analytical relationship between the magnetic spectral index and the level of intermittency has been shown so far. The main novelty of this work is that we show for the first time the analytical relationship between the magnetic spectral index and the level of intermittency by performing a fit on the observational results. Here we will present the continuous variation of the magnetic spectral index as a function of the intermittency level, by using the measurements of the WIND spacecraft in the slow solar wind between 2005 and 2013. More than 42,000 intervals with different levels of the intermittency are selected from 46 slow-wind streams. Our result shows that the magnetic power spectrum between 0.01 Hz and 0.3 Hz in the inertial range gets steeper from $-1.63$ to $-2.01$ as the intermittency level increases from 0 to 20. It will help us to know more detail about the effect of the intermittency on the turbulence cascading process, and will also supply an empirical relation for the theoretical and numerical studies in the future.

The paper is organized as follows. In Section 2, we introduce the data used in this work and the methods applied to find the intermittent structures and to determine both the intermittency level and the spectral index of magnetic fluctuations in the inertial range. In Section 3, we show our observations, including cases and statistical results. In Section 4, we discuss the consequences of our results, and the influence of magnetic compressibility and anisotropy on the results. In Section 5, we summarize this work.

## 2  Data and Methods

We use magnetic field and plasma data measurements both with the time resolution $\Delta t = 3$ s obtained respectively by Magnetic Field Investigation (Lepping et al., 1995) and 3D Plasma Analyzer (3DP, Lin et al., 1995) on board the WIND spacecraft between 2005 and 2013. During this period, the WIND spacecraft was located at the Lagrangian point L1. Here we focus on the slow-wind streams with proton bulk velocity $V_{SW} \leq 450$ km s$^{-1}$, and the data observed within the compression regions that are followed immediately by fast-wind streams are discarded. The compression region is much more complicated and dynamic than the typical slow wind of interest, and it is out of the scope of this work. The plasma data is used here to get the bulk velocity for data selection and to get the proton number density in order to put the magnetic data into Alfvén units. In addition, the plasma data is also used to calculate Alfvénicity for the purpose of revealing the nature of intermittent structures.

From the eight-year observations, we find 46 slow-wind streams. Each of the stream lasts about $2-5$ days. Figure 1 shows one of the selected slow-wind streams observed 12:00:00 UT on 5 Dec 2007 to 00:00:00 UT on 9 Dec 2007. The top three panels are the time variations of the magnetic field (black) and proton velocity (blue) vectors in geocentric solar ecliptic (GSE) coordinates. Panel (d) shows the magnitude of the magnetic field. During the 3.5-day interval, the absolute value of the $x$ component of the proton velocity shown in panel (a) decreases from $\sim$360 km s$^{-1}$ to $\sim$300 km s$^{-1}$. So this interval is out of the compression region, and is not adjacent to fast wind.

A parameter named normalized partial variance of increments $PVI$ is applied to quantitatively analyze the intermittency in the solar wind turbulence following previous studies (Marsch and Tu, 1994; Greco et al., 2008; Osman et al., 2011; Wang et al., 2013). For a component of the magnetic field vector in a given slow-wind stream, the time series of $PVI$ is presented as:

$$PVI_i(t,\tau) = \frac{\delta B_i(t,\tau)}{\sqrt{<|\delta B_i(t,\tau)|^2>}} \tag{1}$$

where $B_i(t)$ is time series of the $i$ component of the magnetic field vector ($i=x,y,z$), $\delta B_i(t) = B_i(t+\tau) - B_i(t)$, and $<...>$ denotes an ensemble average in the given stream. The time lag $\tau$ is selected as 24 s following Wang et al. (2013), corresponding to a spatial separation within the inertial range. In the following we will refer to $PVI_i(t,\tau)$ as $PVI_i(t)$ for simplicity without further specification. Panel (e) of Figure 1 shows the time series of $PVI_z(t)$ for the stream. Many spikes appear in the time series of $PVI_z(t)$, which correspond to large-amplitude fluctuations imbedded in the background turbulence.

In panels (f-h) of Figure 1, we demonstrate the probability distribution function (PDF, solid black curves) of $PVI$ for the three components of the magnetic field, respectively. The dotted curves are standard Gaussian distribution, and they are plotted for easy comparison. The Gaussian distributions are located between the $PVI$ range $[-2,2]$. Beyond this range, the observed distribution curves exhibit long tails, and the tails extend even beyond the plotted range $[-5,5]$. The $PVI$ range $[-2,2]$ will then be used to select intermittent structures. Actually $PVI_z$ can achieve $\pm 10$ as shown in panel (e). So it is clear that the profiles of the PDF for the three components ($PVI_x$, $PVI_y$, and $PVI_z$) all deviate significantly from the Gaussian distribution, and have long tails when the absolute value of $PVI_i$ increases. The long tails of the non-Gaussian PDF profiles indicate the existence of intermittent structures.

We calculate the flatness for each distribution as: $F_i = <(PVI_i(t))^4>/<(PVI_i(t))^2>^2$, where $<...>$ still denotes the ensemble average in the given stream. An empirical rule is that the minimum number N of data points in the time series to be used to accurately calculate moments of order M is $N = 10^{M+1}$ (see Dudok de Wit (2004) for instance). In the case shown in Figure 1, the total number of samplings is 100,800. Since flatness (the fourth-order moment) is considered here, the total number of samplings meets the requirement, which is larger than the minimum number $N = 10^5$. The values of the flatness are marked in the bottom three panels, respectively, as $F_x = 26.1$, $F_y = 30.3$, and $F_z = 43.1$. They are much larger than 3 (characteristic of a standard Gaussian distribution). It again indicates the fluctuations are highly intermittent.

The criterion $|PVI_i(t)| > 2$ is applied for the basic identification of intermittent structures. First, we find the time instants that satisfy at least one of the conditions: $|PVI_x(t)| > 2$, $|PVI_y(t)| > 2$, or $|PVI_z(t)| > 2$. Some of the instants are isolated, and some of them are clustered and continuous. Only if the number of the continuous instants is not smaller than 3, they are chosen for the following study. The rest instants are ignored. A continuous series of the intermittent instants is considered as an intermittent structure. Moreover, if the number of the instants between two adjacent structures is smaller than 3, the two structures and the data points between them are merged together and are seen as one "long-lived" structure. For a given structure, we use $t_B$ and $t_E$ to denote its beginning time and ending time, respectively, and use $(t_E - t_B)/\Delta t$ as the width of the structure in the unit of data point. An interval between $[t_B - 150\ s, t_E + 150\ s]$ is called intermittent interval. After the constrain of $10\%$ data gap, we find $56,398$ intermittent intervals from 46 slow-wind streams between 2005 and 2013.

Figure 2 shows a typical case of an intermittent interval observed by the WIND spacecraft on 2007 Feb 23. Panels (a-c) present the time variations of the three components of the magnetic field vector (black) and the proton velocity vector (gray). The magnetic field data are transformed into Alfvén units (i.e., $\mathbf{B}/\sqrt{\mu_0 m_p \langle n_p \rangle}$ with $\mu_0$ being susceptibility, $m_p$ being proton mass, and $\langle n_p \rangle$ being the average proton number density of the $\sim$5-min interval), so that the fluctuation amplitudes of the magnetic field and the velocity are comparable. Panel (d) shows the time variations of the magnetic field magnitude. Panel (e) shows the time series of $PVI_x(t)$ (purple), $PVI_y(t)$ (yellow), and $PVI_z(t)$ (green), as well as the trace $PVI$ ($I = \sqrt{(PVI_x)^2 + (PVI_y)^2 + (PVI_z)^2}$) (black). The two vertical dotted lines mark the beginning time ($t_B$=01:44:19) and ending time ($t_E$=01:44:34) of the intermittent structure, respectively. We see that between the two vertical lines, $|PVI_z|$ keeps larger than 2 for 15 s (5 data points), which satisfies our criteria of the intermittent structure selection. Accordingly, the width of this intermittent structure obtained from $t_E - t_B$, during which the condition $|PVI_z| > 2$ satisfies, is recorded as 15 s (5 data points).

In the case shown in Figure 2, a very significant jump happens in the $z$ component of the magnetic field between $t_B$ and $t_E$. We also notice that in this case the fluctuation amplitude of the proton velocity is much smaller than the magnetic field (normalized residual energy $\sigma_r = -0.91$), and the fluctuations between the velocity and the magnetic field are not well correlated (correlation coefficient $cc = -0.11$). These characteristics indicate that this case may be associated with magnetic-field directional turning (Tu and Marsch, 1991; Wang et al., 2020). It is convected by the solar wind, and nearly has no velocity fluctuations. Hence it can lead to low normalized residual energy (close to $-1$) and low correlation between $\mathbf{B}$ and $\mathbf{V}$ in the observations.

Next, we determine the intermittency level for each interval. The trace of the normalized partial variance of increments is obtained from $I = \sqrt{(PVI_x)^2 + (PVI_y)^2 + (PVI_z)^2}$. If the maximum $I$ ($I_{max}$) during an intermittent structure (e.g., between the two vertical lines for the case shown in Figure 2) is also the maximum $I$ during the corresponding intermittent interval (e.g., the whole interval for the case shown in Figure 2), this interval will be reserved, and $I_{max}$ is recorded as the intermittency level of this case. Otherwise, the case is eliminated since the energy of the fluctuations during the interval is not dominated by the intermittent structure of interest. In the case shown in Figure 2, we see that $I_{max} = 4.10$ at 01:44:23 is also the maximum $I$ within the plotted interval, so this case satisfies the condition well. In this way, $25,912$ intermittent intervals are reserved for the following analysis.

Then we perform Fast Fourier Transform (FFT) on the magnetic field fluctuations in Alfvén units and obtain the spectral index of the Fourier spectrum in the inertial range. In this procedure, we use high-resolution magnetic field data with $\Delta t_H = 1/11$ s, so that the magnetic spectral index could be more reliable. The high-resolution magnetic field data are first transformed into Alfvén units (i.e., $\mathbf{B}/\sqrt{\mu_0 m_p \langle n_p \rangle}$ with $\langle n_p \rangle$ being the average proton number density of each interval). When putting the magnetic field into Alfvén units, we use one value of proton number density, which corresponds to the ensemble average of proton number density $\langle n_p \rangle$ for each selected interval. By doing so, we avoid the contamination of the noise in density measurements on the magnetic spectral-index value, which would be resulted from using the density value changing every 3 seconds. For a given intermittent interval, the time series of each component of the high-resolution magnetic field data in Alfvén units is Fourier transformed using the FFT method with a simple rectangle window. This method could introduce an extra

discontinuity in the data that will add Fourier power to the magnetic PSD as mentioned by Borovsky (2012) and Borovsky and Burkholder (2020). In subsection 4.4, we apply a linear detrend to the data prior to Fourier transforming following Borovsky (2012), and make a comparison between the two methods. The trace of the magnetic spectral matrix gives the total power spectral density, and the magnetic spectrum is then three-point centered smoothed following Wang et al. (2015). In panel (f) of Figure 2, we plot the magnetic power spectral density (PSD) as a function of the spacecraft frequency ($f$) in log-log space, i.e., $y = log_{10}(PSD)$ versus $x = log_{10}(f)$, as gray curve. We see that by using the high-resolution data, the magnetic spectrum can cover more three decades from $3.3 \times 10^{-3}$ Hz to 5.5 Hz.

It is known that the points of the gray spectrum shown in Figure 2(f) are not uniformly distributed in the logarithm space of $f$. As mentioned in Podesta (2016), in this space the number of data points between two points ($x$ and $x + \Delta x$) increases exponentially with $x$. If a least-squares fit is performed, each point has equal weight. So the fit favors the points in the higher-frequency range since this range contains more points (Podesta, 2016; Borovsky and Burkholder, 2020). In order to avoid this issue, we linearly interpolate the spectral density onto a uniformly spaced grid with $\Delta x = (f_{max} - f_{min})/100$ in the log-log space following Podesta (2016). In panel (f) of Figure 2, the black curve superposed on the original gray spectrum demonstrates the interpolated spectrum.

Then we perform the least squares fit to the interpolated magnetic spectrum to obtain the magnetic spectral index $\alpha_B$ in the log-log space in the inertial range. The least squares fit is performed at the frequency range between 0.01 Hz and 0.3 Hz (between two vertical dotted lines as shown in Figure 2(f)), and at this range the magnetic spectrum can be fitted well by a straight line with a slope of $\alpha_B$. The slope ($\alpha_B$) and its corresponding error ($\Delta_{\alpha_B}$) are obtained and both marked in panel (f) as $\alpha_B = -1.84 \pm 0.04$. We perform the same analysis on all the selected intervals. Then, the cases with the relative fitting error $\Delta_{\alpha_B}/\alpha_B > 5\%$ are eliminated, since the magnetic spectra of them do not have a good power-law shape and cannot be well fitted by a straight line in the log-log space at the frequency range of interest. At last, 24,886 intermittent intervals are reserved for the following statistical analysis to explore the relation between the magnetic spectral index $\alpha_B$ and the intermittency level $I_{max}$.

## 3   Results: Variations of magnetic spectral index versus intermittency level

For the selected 24,886 cases, we first present the joint distribution of their width in units of data points and intermittency level $I_{max}$ in panel (a) of Figure 3. Here, the width in units of data points for an intermittent structure is obtained from $t_E - t_B$, during which the condition $|PVI_i| > 2$ satisfies ($i = x, y,$ or $z$), divided by the time resolution $\Delta t = 3$ s. We see most of the cases have $5 \leq Width < 7$ and $3 \leq I_{max} < 6$. As the width increases, the distribution of $I_{max}$ extends to a wider range. This phenomenon makes the pattern of the joint distribution look like a triangle, which is consistent with Miao et al. (2011). They show in Figure 8 the triangle-like shape of the 2-D distribution in the $\Delta\theta - \tau$ plane, where $\Delta\theta$ and $\tau$ are the deflection angle across current sheet and the width of current sheet, respectively. Panel (b) of Figure 3 shows the probability distribution of the width for the intermittent structures of interest. The width extends from 3 points (9 s) to 20 data points (60 s), and the most probable value is 5 data points (15 s). As the width increases, the probability distribution function first increases immediately

and then decreases gradually. Panel (c) of Figure 3 shows the probability distribution of the intermittency level $I_{max}$. The value of $I_{max}$ extends from about 2 to 15, and the most probable value is 4.5. The profile of the distribution is similar to that of the width.

Another typical intermittent interval is shown in Figure 4 and observed on 2010 Sep 12, but with higher intermittency level $I_{max} = 13.09$. This figure is plotted in the same format as Figure 2. The intermittent structure is marked by the two vertical dotted lines. Between the two vertical lines, the time instants all satisfy at least one of the conditions as mentioned above: $|PVI_x| > 2$, $|PVI_y| > 2$, or $|PVI_z| > 2$. Between $t_B$ and $t_E$, a large jump happens in both the $x$ and $z$ components of the magnetic field. The fluctuations of the proton velocity (in gray) are well correlated with the fluctuations of the magnetic

field (correlation coefficient $cc = 0.97$). However, the fluctuation amplitude of the proton velocity is much smaller than the magnetic field (normalized residual energy $\sigma_r = -0.50$). It indicates that this may be a magnetic-velocity alignment structure (Wang et al., 2020; Wu et al., 2021). Magnetic-velocity alignment structure, of which the generation mechanism remains unclear, is a kind of magnetically dominated structure but with high correlation between magnetic-field fluctuations and velocity fluctuations. For these kinds of structures, the magnetic-field fluctuations are nearly aligned with the velocity fluctuations.

For the case shown in Figure 4, its intermittency level $I_{max}$ is recorded as 13.09, which corresponds to the value of $I$ at 06:28:48. The right panel shows the power spectrum of magnetic field fluctuation obtained from performing FFT on the high-resolution magnetic field data. The original spectrum before interpolation is still plotted in gray, and the uniformly distributed spectrum after interpolation in black is superposed on the gray one. The least squares fit is performed on the interpolated spectrum at the frequency range between 0.01 Hz and 0.3 Hz. The spectral index is obtained as $\alpha_B = -2.01 \pm 0.04$. The small

fitting error indicates that the magnetic spectrum has a good power-law shape. The magnetic spectral index obtained here is very close to $-2$. So it is well consistent with previous theory and observations, which proposed that the discontinuities can produce $f^{-2}$ energy spectrum in the inertial range (Sari and Ness, 1969; Roberts and Goldstein, 1987; Champeney, 1973; Dallas and Alexakis, 2013).

    However, we have seen from the case shown in Figure 2 with $\alpha_B = -1.84 \pm 0.04$ that the discontinuities are not always

related with $-2$ magnetic spectral index in the solar wind observations. The case shown in Figure 2 also have a typical discontinuity imbedded in the background turbulence, but its intermittency level ($I_{max} = 4.10$) is relatively smaller than that shown in Figure 4 ($I_{max} = 13.09$). Correspondingly, the magnetic spectrum of it is shallower. Therefore, it is clear that the intermittency level can affect the spectral index of the magnetic field fluctuations in the inertial range significantly. It is necessary to know the analytical relation between the intermittency level and the magnetic spectral index.

In order to give the continuous variation of the magnetic spectral index as a function of the intermittency level, we also select some "quiet" intervals with $|PVI_i| < 2$. In this procedure, we first cut the data in the 46 slow-wind streams into short intervals with duration of 5 minutes. Then, in each interval we check the maximums of $|PVI_x|$, $|PVI_y|$, and $|PVI_z|$, respectively. If the maximums of them are all smaller than 2, the interval is reserved as a "quiet" interval. During a given interval, the maximum of the trace $I = \sqrt{(PVI_x)^2 + (PVI_y)^2 + (PVI_z)^2}$ is recorded as the "intermittency level" ($I_{max}$), although it may

not be intermittent at all. The magnetic spectral indices of them are also obtained by using the method mentioned above. Subsequently, we find 17,386 quiet cases for the following study.

Figure 5 shows a typical quiet interval with very low intermittency level $I_{max} = 1.44$ in the same format as Figure 2. The magnetic power spectrum is much shallower than that of the intermittent intervals with relatively higher intermittency levels shown in Figure 2 and Figure 4. The magnetic spectral index comes out to be $-1.65 \pm 0.04$. It seems to be close to the Kolmogorov scaling $f^{-5/3}$. We check the Alfvénicity of this case, and find that it is not an Alfvénic interval with low normalized cross helicity $\sigma_c = 0.34$ and low Alfvén ratio $\gamma_A = 0.47$. It's worth noting that for an Alfvénic interval, if the magnetic spectrum scales as $f^{-5/3}$, an intermittency correction could be considered.

The lower panel of Figure 6 shows the joint distribution of $I_{max}$ and $\alpha_B$ for the selected 42,272 intervals. The $x$ axis corresponding to the intermittency level $I_{max}$ in the range $[0, 20]$ is divided into 20 bins. The $y$ axis corresponding to the magnetic spectral index $\alpha_B$ in the range $[-2.5, -1.2]$ is divided into 13 bins. For a given pixel, the color denotes the number of the cases normalized by the maximum number of the pixels among the corresponding $I_{max}$ bin. Thus, in each column, the pixel with the largest amount of cases is colored in red, corresponding to 1. The maximum number of each column versus $I_{max}$ is also shown in the upper panel of Figure 6. In order to guarantee that there are enough cases used for statistics, the pixels containing no more than 10 cases are ignored. So the pixels in black contains the smallest amount of cases, but the number of the cases is still larger than 10. If we focus on the pixels in red, we notice that when the intermittency level $I_{max}$ increases, the magnetic spectral index $\alpha_B$ has a very clear decreasing trend from $\sim -1.6$ to $\sim -2$. The gray solid circles show the average $\alpha_B$ in each $I_{max}$ bin as a function of $I_{max}$, and the dotted gray lines represent the upper/lower quartiles. It is found that as $I_{max}$ increases from 0.5 to 3.5, the magnetic power spectrum gets steeper quickly from $f^{-1.63^{+0.09}_{-0.12}}$ to $f^{-1.84^{+0.14}_{-0.11}}$. When $I_{max}$ increases from 4.5 to 15.5, the magnetic power spectrum gets steeper slowly from $f^{-1.86^{+0.14}_{-0.11}}$ to $f^{-1.99^{+0.09}_{-0.11}}$. As $I_{max} > 16$, the magnetic spectral index keeps close to $-2$.

The observed variation of the magnetic spectral index $\alpha_B$ versus the intermittency level $I_{max}$ can be well fitted by an exponential function. In Figure 6, the black curve corresponding to $\alpha_B = 0.4 \exp(-I_{max}/5) - 2.02$ shows the fitting result. This empirical relation supplies the continuous variation of the spectral index $\alpha_B$ of magnetic field in the inertial range as a function of the intermittency level $I_{max}$. The empirical relation tells us that when $I_{max}$ is small, and the fluctuations of the magnetic field could be considered as randomly distributed, the magnetic spectral index in the inertial range will be close to $-1.6$. As the fluctuations get intermittent, the magnetic spectrum becomes steeper gradually until $\sim f^{-2}$.

## 4 Discussion

Our result confirms the idea that the intermittent structures have significant influence on the magnetic spectral index and often make the spectra become steeper (Siscoe et al., 1968; Burlaga, 1968; Salem et al., 2007, 2009). It is generally acknowledged that the time series of discontinuities produce $f^{-2}$ energy spectrum in the inertial range. Later, people found that the discontinuities can also produce shallower magnetic spectra (Li et al., 2011; Borovsky, 2010). In previous studies, the discontinuities in the solar wind have been identified mainly as rotational discontinuities (e.g., Neugebauer et al., 1984; Tsurutani and Ho, 1999; Wang et al., 2013; Liu et al., 2021). However, in some other studies, the discontinuities have been identified mainly as tangential discontinuities, depending on the different techniques used for data analysis (e.g., Horbury et al., 2001; Knetter et al., 2004;

Riazantseva et al., 2005). Here, we find from the continuous relation that the $f^{-2}$ scaling could be produced if the intermittency level of the structure imbedded in the turbulence is high enough, i.e., $I_{max} > 15$ for the cases studied in this work. We have checked about whether the intermittency level $I_{max}$ could be biased by the anisotropy of fluctuations. It is found that the intermittency level $I_{max}$ appears to be not dependent on the direction of the predominant fluctuations (figure not shown here, since it is similar as Figure 9 shown below).

Our result is also consistent with the radial evolution trend of intermittency and magnetic spectral index in the solar wind. The evolution of intermittency with distance from the Sun can be explained on the basis of the interplay between coherent (intermittent) structures and Alfvénic fluctuations. Intermittent events advected by the wind are increasingly exposed as the Alfvénic fluctuations are depleted with the heliocentric distance (see, for instance, Bruno et al. (2003)). By using the observations from Parker Solar Probe, people also found that there is a clear transition for the magnetic spectral index in the inertial

range as the radial distance from the Sun increases (Chen et al., 2020). When $r \approx 0.17$ au, the magnetic spectral index is close to $-3/2$. When $r \approx 0.6$ au, the magnetic spectrum becomes steeper as $\alpha \approx -5/3$. These observational results indicate that when $r$ increases, the solar wind turbulence becomes more intermittent, and the magnetic spectrum gets steeper. The variation trend of the magnetic spectral index versus the intermittency is confirmed by our observations. Recently, there are several papers on the scaling properties and intermittency levels with Parker Solar Probe (e.g., Alberti et al., 2020; Cuesta et al., 2022;

Sioulas et al., 2022).

  We also notice that for the cases with very low intermittency level $0 < I_{max} < 1$, the magnetic spectral index of the intervals is between $-1.62$ and $-1.69$, which is close to the Kolmogorov scaling. This is different from Li et al. (2011), who found that the current-sheet free periods show $f^{-3/2}$ Iroshnikov-Kraichnan scaling. We see from Figure 6 that some of the low-intermittency-level cases can also produce the $f^{-3/2}$ scaling, but the number of the cases with $-1.6 < \alpha_B < -1.4$ only account

for 20% of all the cases with $I_{max} < 1$. The differences between this work and Li et al. (2011) include: they focus on the $\sim$1-day Ulysses data at about 5 AU, while we use the 5-minute WIND data at about 1 AU. In addition, the frequency range for the fittings is $[10^{-3}, 10^{-1}]$ Hz in Li et al. (2011) and $[0.01, 0.3]$ Hz here.

  The $f^{-2}$ scaling has been reported for parallel-sampling magnetic fluctuations in many previous studies associated with magnetic spectral anisotropy (Horbury et al., 2008; Podesta, 2009; Luo and Wu, 2010; Chen et al., 2011; Forman et al., 2011;

Wicks et al., 2011). Wang et al. (2014) found that after removing the intermittency, the magnetic spectrum in the parallel direction becomes shallower from $f^{-2}$ to $f^{-5/3}$. However, the question about how the intermittency affect the anisotropy of the magnetic spectral index remains unclear. In the future, we might try to check the intermittency level of the parallel-sampling data to see if the steep spectrum in the parallel direction is related to high intermittency level or not.

  The intermittency in many theoretical models are also found to steepen the inertial-range power spectrum of turbulence. For

example, a multi-fractal model developed by She and Leveque (1994) (SL model) gave intermittency correction to the Kolmogorov law (Kolmogorov, 1941), and predicted an energy spectrum $E(k) \approx k^{-5/3-0.03}$ for fluids. Carbone (1993) presented a magnetohydrodynamic (MHD) cascade model and found the intermittency modification to the Kraichnan theory. Politano and Pouquet (1995) extended the SL model to the MHD case, and the energy spectrum was obtained as $E(k) \approx k^{-3/2-0.04}$. Boldyrev et al. (2002) predicted for the velocity spectrum $E(k) \approx k^{-1.74}$ from an analytical study of driven supersonic MHD

turbulence. Recently, Chandran et al. (2015) found that when considering scale-dependent dynamic alignment, the power spectrum of the intermittent turbulence flattens. However, there seems no conclusion about which model is the most appropriate one to describe the solar wind turbulence. According to the observational result shown in this work in the slow-wind streams, we obtain the empirical relation between the magnetic spectral index $\alpha_B$ and the intermittency level $I_{max}$. The relation will supply observational basis for theoretical studies of the intermittent turbulence, and will help improve the turbulence theory related to the slow solar wind.

### 4.1 Influence of magnetic compressibility

Besides the intermittency, the magnetic spectral index has been reported to also depend on the level of magnetic compressibility. The magnetic compressibility was defined as the ratio between the variance of the magnetic field magnitude fluctuations and the variance matrix trace of the fluctuations, i.e., $c_b = \sigma_{|B|}^2 \, / \, \sum_{i=x,y,z} \sigma_{B_i}^2$ (Bavassano et al., 1982; Telloni et al., 2019; Wang et al., 2020). Here, in order to take into account of the influence of the magnetic compressibility on the shape of the magnetic spectrum, we also calculate the magnetic compressibility of all the 24,886 intermittent intervals and 17,386 quiet intervals in the 46 slow-wind streams.

The lower panel of Figure 7 shows the joint distribution of the magnetic compressibility $c_b$ and $\alpha_B$ for the selected 24,886 intermittent intervals in the same format as Figure 6. The $x$ axis corresponding to the magnetic compressibility $c_b$ in the range $[0, 0.5]$ is divided into 20 bins. The $y$ axis corresponding to the magnetic spectral index $\alpha_B$ in the range $[-2.5, -1.2]$ is still divided into 13 bins. For a given pixel, the color also denotes the number of the cases normalized by the maximum number of the pixels among the corresponding $c_b$ bin. The maximum number of each column versus $c_b$ is also shown in the upper panel of Figure 7. The pixels containing no more than 10 cases are ignored. When we focus on the pixels in red, we notice that when $c_b$ increases, the magnetic spectral index $\alpha_B$ keeps nearly constant. The gray solid circles show the average $\alpha_B$ in each $c_b$ bin, and the two dotted gray lines represent the upper and lower quartiles, respectively. It is found that for the selected intermittent intervals, as the magnetic compressibility $c_b$ increases from 0 to 0.5, the average slope of the magnetic spectrum in the inertial range varies between $[-1.90, -1.80]$, and there is no systematic trend. This result could indicate that for the intermittent cases, the magnetic compressibility does not have significant influence on the magnetic spectral index in the slow-wind streams of interest.

The same analysis is performed on the selected 17,386 quiet intervals. The result is shown in Figure 8. Figure 8 is plotted in the same format as Figure 7. When we focus on the most probably value of $\alpha_B$ in each $c_b$ bin, i.e., the pixels in red, we can find that no clear trend appears, neither. The gray solid circles and the two dotted gray lines represent the average $\alpha_B$ in each $c_b$ bin and the upper/lower quartiles, respectively. When $c_b$ increases from 0 to 0.5, the magnetic spectral index changes slightly from $-1.76 \pm 0.14$ to $-1.70 \pm 0.10$. The result indicates that for the quiet cases in the slow-wind streams of interest, the magnetic compressibility does not significantly affect the magnetic spectral index, neither.

## 4.2 Influence of anisotropy of magnetic field fluctuations

Since different spectral indices are observed if looking along different directions with respect to the mean field as mentioned in the introduction, it is necessary to reveal how the presented results shown in Figure 6 could be biased by the anisotropy of magnetic field fluctuations. We then perform a check to see if the spectral slope is dependent on the predominance of fluctuations along a specific direction. Here the direction of the predominant fluctuations is indicated by the maximum variance (**L**) direction, which is obtained from the Minimum Variance Analysis (Sonnerup and Cahill, 1967). We show in Figure 9 the variations of the magnetic spectral index as a function of the angle between **L** and $i$ direction ($\theta_{Li}$) (where $i$ denotes the $x$-axis, $y$-axis, and $z$-axis of geocentric solar ecliptic coordinates), along with the variations of the spectral index versus the angle between **L** and the mean magnetic field direction of each interval ($\theta_{LB}$).

Panel(a2) of Figure 9 shows the variation of the magnetic spectral index $\alpha_B$ as a function of $\theta_{LX}$. The angle $\theta_{LX} \sim 0°$ means that the predominant fluctuations of the intermittent structure mainly focus on the $x$ direction, while $\theta_{LX} \sim 90°$ means that they focus on the plane perpendicular to the $x$ direction. Only 79% of the selected intervals with $\lambda_1/\lambda_2 > 3$ are remained for the analysis, where $\lambda_1$ and $\lambda_2$ are the eigenvalues corresponding to the maximum variance direction and the intermediate variance direction, respectively. This condition guarantees that the maximum variation direction is determined precisely, and the fluctuations in the **L** direction are distinctly dominant in each interval. Panels (a1) and (a2) are plotted in the similar format as Figure 6. For a given pixel in panel (a2), the color denotes the number of cases normalized by the maximum number among the corresponding $\theta_{LX}$ bin, and the maximum number of each bin is shown in panel (a1). The gray solid circles represent the average $\alpha_B$ in each $\theta_{LX}$ bin. The dotted gray lines represent the upper/lower quartiles. The gray solid circles show that there is a slight decreasing trend for the average spectral index $\alpha_B$ (from $-1.76$ to $-1.86$) as $\theta_{LX}$ increase from $0°$ to $90°$. However, if we consider the quartiles (i.e., from $-1.76^{+0.14}_{-0.10}$ to $-1.86^{+0.13}_{-0.14}$), the slight trend is nearly negligible. Therefore, the magnetic spectral indices of the intervals with the predominant fluctuations parallel or perpendicular to the $x$ direction are not significantly different.

Figure 9bcd show the variation of the magnetic spectral index as a function of $\theta_{LY}$, $\theta_{LZ}$, and $\theta_{LB}$. A slight increasing trend (from $-1.89^{+0.17}_{-0.17}$ to $-1.84^{+0.14}_{-0.13}$) appears in panel (b2), but the trend is not significant, neither, considering the errors. In panel (c2), the average $\alpha_B$ (gray solid circles) nearly keeps constant at $-1.85$. In panel (d2), the average $\alpha_B$ (gray solid circles) varies with $\theta_{LB}$, and no clear trend exists.

According to the results presented in the panels of Figure 9, we suggest that the influence of the anisotropy of the predominant fluctuations on the magnetic spectral index is not as significant as the influence of the intermittency level $I_{max}$ on the index (when $I_{max}$ increases from 0 to 20, $\alpha_B$ decreases from $-1.63$ to $-2.01$).

## 4.3 Coincidence between intermittency level and multifractal width

As shown in literature (e.g., Frisch (1995); Veltri and Mangeney (1999); Salem et al. (2009)), intermittency is strictly related to multifractality that is measured by looking at the high-order scaling properties. Therefore, it is necessary to check if $I_{max}$ used

here is consistent with multifractal indicators of intermittency, such as the multifractal width introduced in a series of work by Macek, Wawrzaszek et al. .

The multifractal properties can be described by the multifractal singularity spectrum of the observed time sequence. The width of the spectrum represents the extent of multifractality. Here we estimate the multifractal singularity spectrum of the magnetic field fluctuations by using the classical approach following previous studies (Paladin and Vulpiani, 1987; Macek et al., 2005; Macek and Wawrzaszek, 2009; Macek et al., 2012; Marsch et al., 1996; Burlaga, 1991a; Burlaga et al., 2006; Sorriso-Valvo et al., 2017). For each selected interval, we perform the multifractal analysis on the time sequence of the magnetic field

fluctuations in the maximum variance direction ($B_L(t)$) with high time resolution of $\Delta t_H = 1/11$ s. The increment of $B_L(t)$ is $\Delta B_L(t) = |B_L(t+dt) - B_L(t)|$, where $dt = 10s$ belongs to the inertial range. The time series $\Delta B_L(i)$ ($i = 1, 2, ..., N$, with $N = T/\Delta t_H$ and $T$ being the duration of each interval) is divided into subsets of variable scale $\Delta s$, with $j = 1, 2, ..., M$ ($M = T/\Delta s$). A logarithmically spaced range of eight time scales $10/11$ s $< \Delta s <$ 150 s is used. For each subset, the generalized probability measure is defined as

$$\mu_j(\Delta s) = \frac{\sum_{i=(j-1)\Delta s+1}^{j\Delta s} |\Delta B_L(i)|}{\sum_{i=1,N} |\Delta B_L(i)|}. \tag{2}$$

For a given $q$, we calculate the $q-$order total probability measure, and it scales as

$$\chi_q(\Delta s) = \sum_{j=1}^{M} \Delta s |\mu_j(\Delta s)|^q \propto (\Delta s)^{\tau_q}, \tag{3}$$

where $q \in [-5, 5]$ with a step $dq = 1/3$ (similar to Sorriso-Valvo et al. (2017)). The scaling exponents $\tau_q$ is obtained by performing a linear fit on the log-log plot of $\chi_q(\Delta s)$ versus $\Delta s$ in the inertial range $[8s, 100s]$. We then obtain the singularity

spectrum from $f(\alpha) = q\alpha_q - \tau_q$ and $\alpha_q = d\tau_q/dq$ (Halsey et al., 1986). The left panel of Figure 10 presents the variations of $f(\alpha)$ versus $\alpha$, with red for the intermittent interval shown in Figure 4 ($I_{max} = 13.09$), black for the intermittent interval shown in Figure 2 ($I_{max} = 4.10$), and blue for the quiet interval shown in Figure 5 ($I_{max} = 1.44$). The dots and solid lines denote the observational results and cubic polynomial fitting to them, respectively.

A quantitative description of the degree of multifractality is the width of the singularity spectrum $\Delta \alpha = \alpha_{max} - \alpha_{min}$. We

estimate $\alpha_{min}$ and $\alpha_{max}$ by fitting the observed values of $(\alpha, f(\alpha))$ with the cubic polynomial and extrapolating to $f(\alpha) = 0$ as shown in the left panel of Figure 10. We find that the multifractal widths of the two intermittent intervals ($\Delta \alpha = 1.19$ in red and $\Delta \alpha = 1.16$ in black) are both much larger than that of the quiet interval ($\Delta \alpha = 0.81$ in blue). Moreover, the intermittent interval with higher level of intermittency ($I_{max} = 13.09$) also corresponds to wider singularity spectrum $\Delta \alpha = 1.19$ in red, comparing to the black one ($I_{max} = 4.10$ and $\Delta \alpha = 1.16$).

In the right panel of Figure 10, we show the statistical results of the multifractal width $\Delta \alpha$ versus the level of intermittency $I_{max}$ for the 33,261 intervals with $\lambda_1/\lambda_2 > 3$ as mentioned in subsection 4.2. They are found to be positively correlated. When $I_{max} < 3$, the multifractal width $\Delta \alpha$ rapidly increases from 0.8 to 1.05. When $I_{max} > 3$, $\Delta \alpha$ increases slowly from 1.05 to $\sim 1.2$. Accordingly, we suggest that, to some extent, the multifractal width $\Delta \alpha$ and the level of intermittency $I_{max}$ coincide with each other.

## 4.4 Linear detrending to data prior to FFT

When performing the FFT on the components of magnetic field data, we use a simple rectangle window (hereinafter referred to as "no data preprocessing" method). This method could introduce an extra discontinuity in the data that will add Fourier power to the magnetic PSD as mentioned by Borovsky (2012) and Borovsky and Burkholder (2020). Following Borovsky (2012), we try applying a linear detrend to each data interval prior to Fourier transforming (hereinafter referred to as "linear detrending preparation" method), and compare the result with that in Figure 6 obtained from "no data preprocessing" method.

Figure 11 presents the joint distribution of intermittency level $I_{max}$ and magnetic spectral index $\alpha_B$ obtained from "linear detrending preparation" method plotted in the same format as the lower panel of Figure 6. The analytical relationship $\alpha_B = 0.4\exp(-I_{max}/5) - 2.02$ adopted from Figure 6 is superposed on the figure as black curve for easier comparison. It is clear that when $I_{max} > 12$, the black curve coincides with the averaged magnetic spectral indices $\alpha_B$ (gray dots) well. However, when $I_{max} < 12$, the averaged magnetic spectral indices $\alpha_B$ (gray dots) obtained from "linear detrending preparation" method appear to be larger than that obtained from "no data preprocessing" method denoted by the black curve. The differences between them are about $0.01 - 0.06$. This is consistent with Borovsky (2012), which mentioned that the "no data preprocessing" method leads to spectral indices slightly steeper. When looking at the upper/lower quartiles, we notice that the distribution of $\alpha_B$ in a $I_{max}$ bin obtained from "linear detrending preparation" method (e.g., $\alpha_B = -1.90^{+0.15}_{-0.14}$ at $I_{max} = 8.5$) is slightly wider than that obtained from "no data preprocessing" method (e.g., $\alpha_B = -1.93^{+0.13}_{-0.12}$ at $I_{max} = 8.5$). The wider distribution for the "linear detrending preparation" method is also consistent with Borovsky (2012). Accordingly, we suggest that when using different data preprocessing methods, the magnetic spectral index slightly changes, but our results about the trend of the magnetic spectral index $\alpha_B$ versus the intermittency level $I_{max}$ and the contribution of the intermittency on the magnetic spectra are robust.

## 5  Conclusions

In this paper, we present for the first time the analytical relation between the magnetic spectral index $\alpha_B$ in the inertial range and the level of intermittency $I_{max}$ at the time scale of $\tau = 24$ s in the slow solar wind. The data from the WIND spacecraft observations between 2005 and 2013 are used for analysis. We examine 56,398 intermittent structures preliminarily by using the criterion $|PVI_i| > 2$ ($i = x, y$, or $z$), with $t_B$ and $t_E$ being the beginning and ending instants of a structure, respectively. However, for more than half of them, the maximum $I$ ($I_{max}$) during $[t_B, t_E]$ (as marked by the two vertical dotted lines in Figure 2) is not the maximum $I$ during the corresponding plotted interval $[t_B - 150s, t_E + 150s]$ (as the whole plotted interval in Figure 2). It means that outside $[t_B, t_E]$, there exist some other structures with even higher level of intermittency during the interval $[t_B - 150s, t_E + 150s]$. We eliminate this kind of intervals, during which the energy of the fluctuations is not dominated by the intermittent structure imbedded in the center of it. In this way, we avoid the duplicate selection of the cases, and also guarantee that both the intermittency level $I_{max}$ and the magnetic spectral index $\alpha_B$ are closely related to the intermittent structure imbedded in the middle of each interval. Then we obtain 25,912 intermittent intervals. Subsequently, the

cases with higher fitting error of the magnetic power spectra ($\Delta_{\alpha_B}/\alpha_B > 5\%$) are eliminated, and 24,886 intermittent intervals are reserved for the statistical analysis.

At last, we select 24,886 intermittent intervals and 17,386 quiet intervals from 46 slow-wind streams. Each intermittent interval lasts about $5 \sim 6$ minutes with a dominant intermittent structure imbedded in the center of it. The maximum $I$ ($I_{max}$) of an intermittent structure is recorded as the intermittency level of the corresponding interval. The magnetic trace power spectrum of each interval is obtained by performing FFT on the high-resolution magnetic field data with $\Delta t_H = 1/11$ s, and is then linearly interpolated onto a uniformly spaced grid in the log-log space. The magnetic spectral index $\alpha_B$ is obtained by performing the least squares fit on the interpolated spectrum between 0.01 Hz and 0.3 Hz in the inertial range. The selected intervals all have relatively low fitting errors ($\Delta_{\alpha_B}/\alpha_B \leq 5\%$), indicating that the magnetic power spectra of them have good power-law shape.

The observed variation of the averaged spectral index $\alpha_B$ as a function of the intermittency level $I_{max}$ is presented in the lower panel of Figure 6 as gray solid circles. When $I_{max}$ increases from 0.5 to 15.5, the magnetic power spectrum gets steeper, and the averaged magnetic spectral index $\alpha_B$ decreases from $-1.63^{+0.09}_{-0.12}$ to $-1.99^{+0.09}_{-0.11}$. We also find that the averaged magnetic spectral index $\alpha_B$ changes more quickly at $I_{max} \leq 3.5$ than at $3.5 < I_{max} \leq 15.5$. When $I_{max}$ gets larger, the magnetic spectral index stops decreasing and keeps nearly constant at $\alpha_B \approx -2$. However, the dependence of the magnetic spectral index on the magnetic compressibility seems to be not significant as shown in Figure 7 and Figure 8.

According to the observational result, an empirical relation is built up between the magnetic spectral index $\alpha_B$ and the intermittency level $I_{max}$ as $\alpha_B = 0.4 \exp(-I_{max}/5) - 2.02$. The empirical relation is illustrated as black curve in the lower panel of Figure 6. It gives the continuous variation of the magnetic spectral index $\alpha_B$ as a function of the intermittency level $I_{max}$. This relation will help people to easily estimate the contribution of the intermittency level on the magnetic spectral index, which implies the nature of the cascading process happening in the turbulence. It also supplies observational constraint for numerical studies related to the intermittency and spectral analysis about the solar wind turbulence. From the aspect of theoretical study, the relation will also help improve the turbulence theory that contains intermittent structures.

We also check the sensitivity of the results based on the choice of the threshold for identifying an intermittent interval. The threshold is changed from the original PVI range $[-2, 2]$ into two new ranges $[-1, 1]$ and $[-3, 3]$. The results are shown in Figure 12. The left panels and right panels correspond to the thresholds $[-1, 1]$ and $[-3, 3]$ for identifying an intermittent interval, respectively. They are plotted in the same format as Figure 6. The black curves in the lower two panels are both the exponential function $\alpha_B = 0.4 \exp(-I_{max}/5) - 2.02$, which is adopted from Figure 6. It is found that the black curve obtained from the original threshold $[-2, 2]$ can still match the new results well. Therefore, our result shown in Figure 6 is robust, and is not sensitive to the choice of the threshold for identifying intermittent intervals.

Moreover, Sari and Ness (1969) has mentioned that "The only change in the spectra for intervals containing a different number of discontinuities, or of discontinuities of differing magnitude, should be in the power levels, and not in the general spectral shape." Based on the high-resolution data and sufficient samples observed by the WIND spacecraft, our result here provide the observational evidence that the magnetic spectral shape (i.e., the spectral index in the inertial range) actually changes when the intermittency level of interval is different. So, when people try to study the cascading process and evolution

of the solar wind turbulence, it is very necessary to consider the effect of the intermittency level. In the future, we will also investigate the influence of the number of intermittent structures on the magnetic spectral shape. Additionally, it will be also interesting to know the physical nature of these intermittent structures found in the slow-wind streams, and to compare the

460 result with that in the fast-wind streams (Wang et al., 2013).

*Data availability.* WIND data are downloaded from SPDF (http://spdf.gsfc.nasa.gov). The magnetic field data used in this work include 3s-resolution (WI_H0_MFI) and high-resolution (WI_H2_MFI) data measured by Magnetic Fields Investigation between 2005 and 2013. The plasma data used here include 3s-resolution ion moments (WI_PM_3DP) measured by 3D Plasma Analyzer between 2005 and 2013.

*Author contributions.* XW had the main responsibility of the data analysis and writing of the article. XHF also participated in the data
analysis. YXW, HHW and LZ participated in the discussion and interpretation of the results, as well as editing of the manuscript text.

*Competing interests.* None of the authors has any competing interests.

*Acknowledgements.* This work at Beihang University is supported by the National Natural Science Foundation of China under contract Nos. 41874199, 41974198, and 41504130. X. Wang is also supported by the Fundamental Research Funds for the Central Universities of China (KG16152401, KG16159701). This work is also supported by the B-type Strategic Priority Program of the Chinese Academy of Sciences
(grant No. XDB41000000) and the pre-research projects on Civil Aerospace Technologies No. D020103 and D020105 funded by China's National Space Administration (CNSA).

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

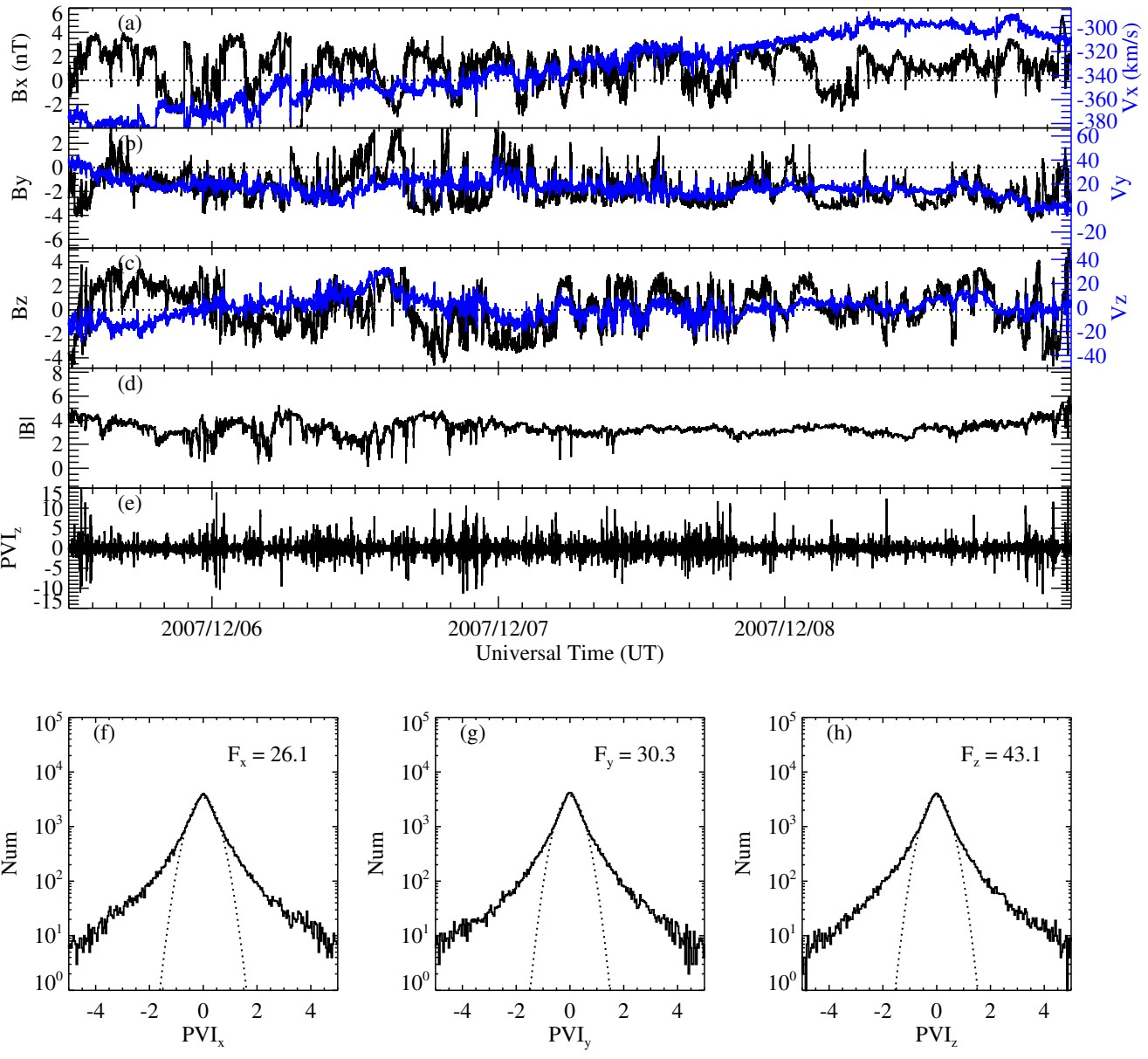

**Figure 1.** A typical slow-wind stream observed from 12:00:00 UT on 5 Dec 2007 to 00:00:00 UT on 9 Dec 2007 by the WIND spacecraft at the L1 point. (a-c) Time variations of the three components of magnetic field vector (black) and proton velocity vector (blue) in the GSE coordinates. Horizontal dotted lines correspond to 0 nT. (d) Magnetic field magnitude. (e) Normalized partial variance of increments ($PVI$) for the $z$ component of the magnetic field vector at the time scale of $\tau = 24$ s. (f-h) Probability distribution function (PDF, solid black curves) of $PVI$ for the three components of magnetic field, respectively. Flatness $F_i$ $(i = x, y, z)$ of each distribution is marked in each panel. Dotted curves denote standard Gaussian distribution.

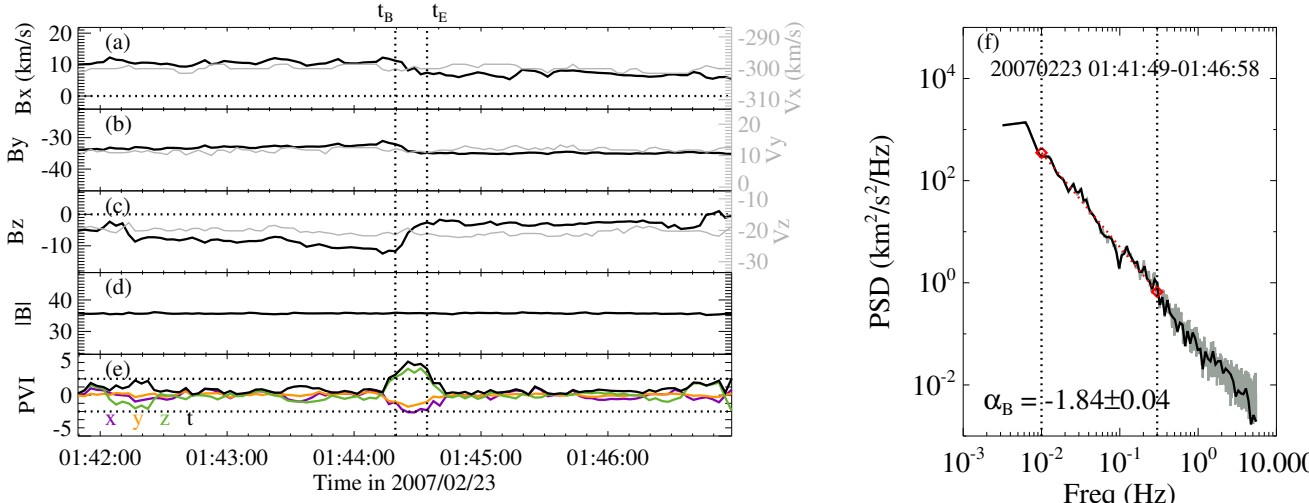

**Figure 2.** A typical case of intermittent interval observed by the WIND spacecraft at 01:41:49−01:46:58 on 2007 Feb 23. (a-c) Time variations of the magnetic field vector (black) and the proton velocity vector (gray) in the GSE coordinates. The magnetic field is plotted in Alfvén units (i.e., $\mathbf{B}/\sqrt{\mu_0 m_p \langle n_p \rangle}$ with $\mu_0$ being susceptibility, $m_p$ being proton mass, and $\langle n_p \rangle$ being the average proton number density of this interval). (d) Magnetic field magnitude in Alfvén units. (e) Normalized partial variance of increments ($PVI$) for the magnetic field vector at the time scale of $\tau = 24$ s, with $PVI_x$ in purple, $PVI_y$ in orange, $PVI_z$ in green, and matrix trace of $PVI$ in black. The two horizontal lines correspond to $|PVI| = 2$ that used to search for the intermittent structure. The two vertical dotted lines mark the beginning time ($t_B$) and the ending time ($t_E$) of the intermittent structure, respectively. (f) Spacecraft-frame trace power spectra of magnetic field fluctuations. The gray curve corresponds to the magnetic power spectrum by performing FFT on the 1/11-s-resolution magnetic field data in Alfvén units obtained still from $\mathbf{B}/\sqrt{\mu_0 m_p \langle n_p \rangle}$ with a simple rectangle window. The black curve superposed on the gray one corresponds to the uniformly distributed spectrum after interpolation. The magnetic spectral index and its uncertainty shown are obtained from applying a least-squares fit to the interpolated spectrum, resulting in the straight line (red dotted line) over the frequency range from 0.01 Hz to 0.3 Hz (between the two vertical dotted lines).

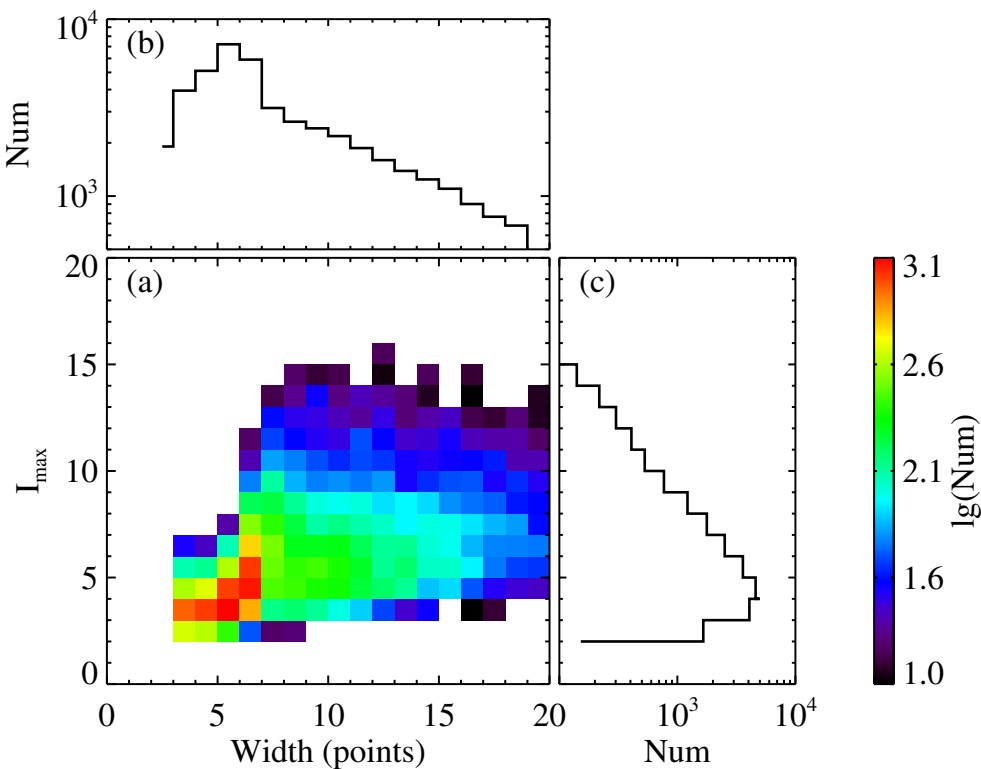

**Figure 3.** (a) Joint distribution of width in units of data points and intermittency level ($I_{max}$) for the selected 24,886 intermittent structures. The width in units of data points for an intermittent structure is obtained from $t_E - t_B$, during which the condition $|PVI_i| > 2$ satisfies ($i = x, y,$ or $z$), divided by the time resolution $\Delta t = 3$ s. The pixels containing no more than 10 cases are ignored. (b) Probability distribution of the width. (c) Probability distribution of the intermittency level ($I_{max}$).

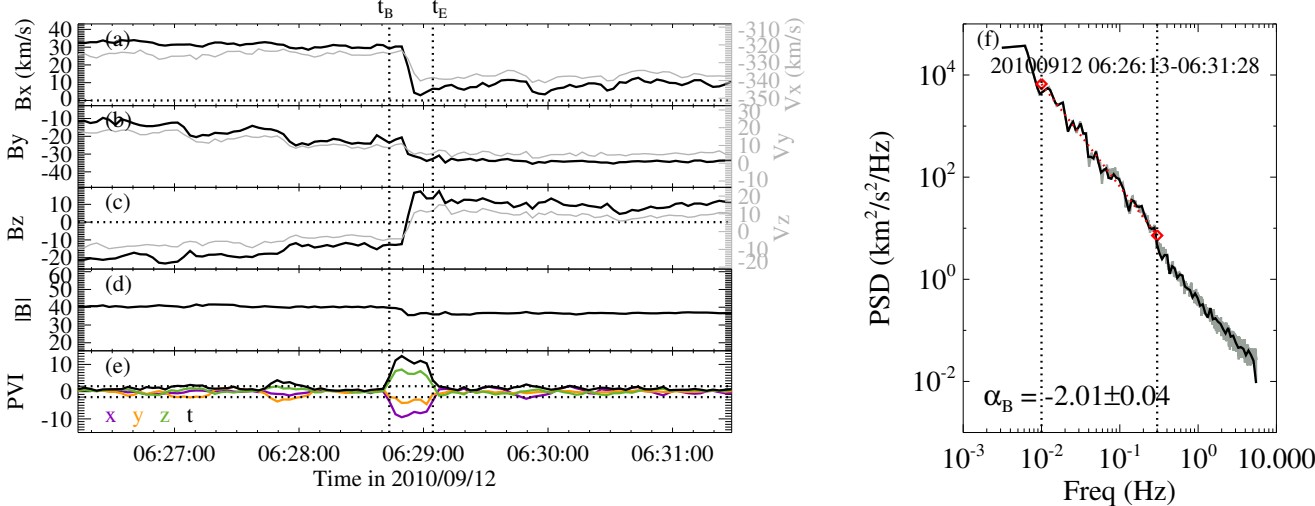

**Figure 4.** A typical case of intermittent interval observed by the WIND spacecraft at 06:26:13−06:31:28 on 2010 Sep 12 in the same format as Figure 2.

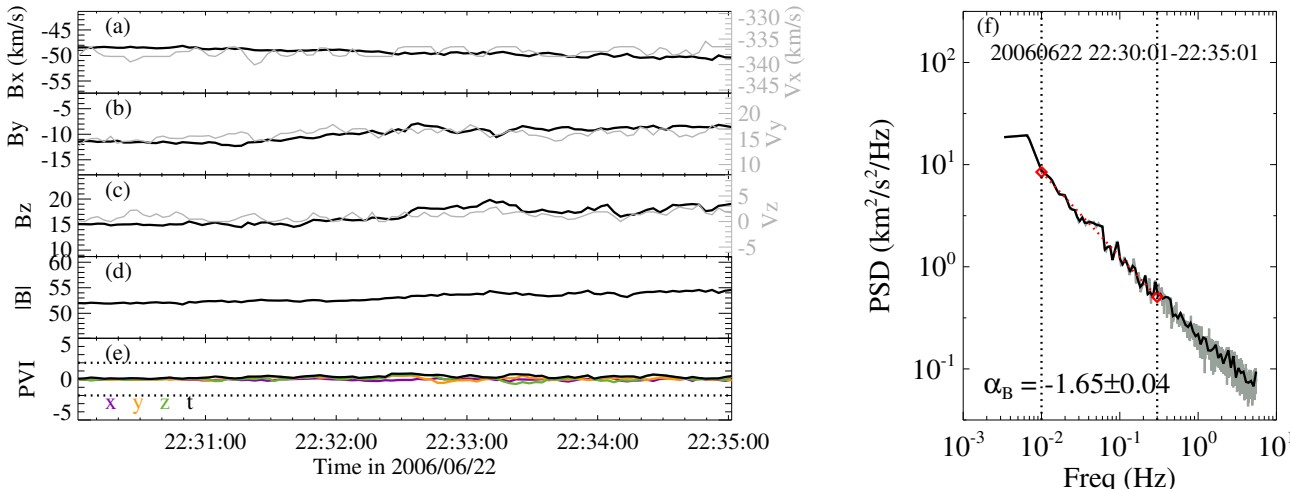

**Figure 5.** A typical case of quiet interval observed by the WIND spacecraft at 22:30:01−22:35:01 on 2006 Jun 22 in the same format as Figure 2. The fluctuation amplitude of the proton velocity (gray curves in panels (a)(b)(c)) is very close to the instrument noise level (2 km s$^{-1}$) in the 3DP velocity observations (Wicks et al., 2013a). Thus, the variations of the proton velocity look noisy here.

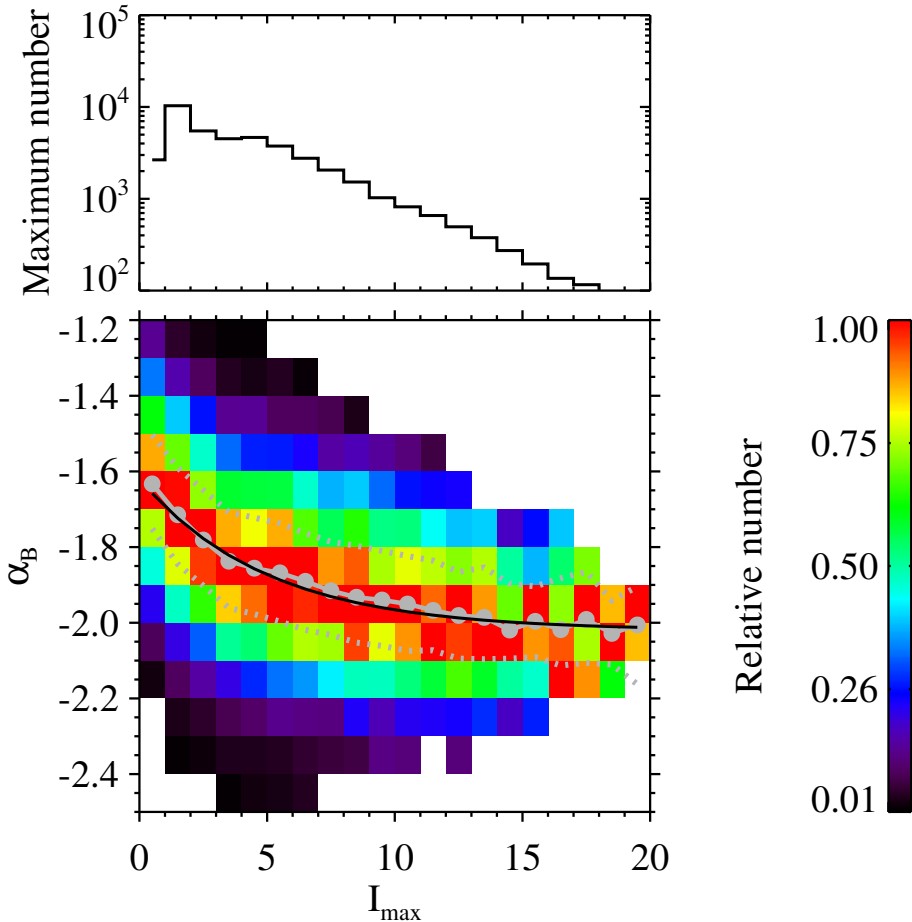

**Figure 6.** Joint distribution of intermittency level $I_{max}$ and magnetic spectral index $\alpha_B$ for the selected 42,272 intervals. The bin width of $I_{max}$ is 1, and the bin width of $\alpha_B$ is 0.1. For a given pixel, the color denotes relative number, which is the number of the cases normalized by the maximum number among the corresponding column ($I_{max}$ bin). the maximum number of each bin is shown in the upper panel. The pixels containing no more than 10 cases are ignored. The gray solid circles represent the average $\alpha_B$ in each $I_{max}$ bin. The dotted gray lines represent the upper/lower quartiles. The black curve, corresponding to the exponential function $\alpha_B = 0.4\exp(-I_{max}/5) - 2.02$, represents the fitting result to the gray solid circles.

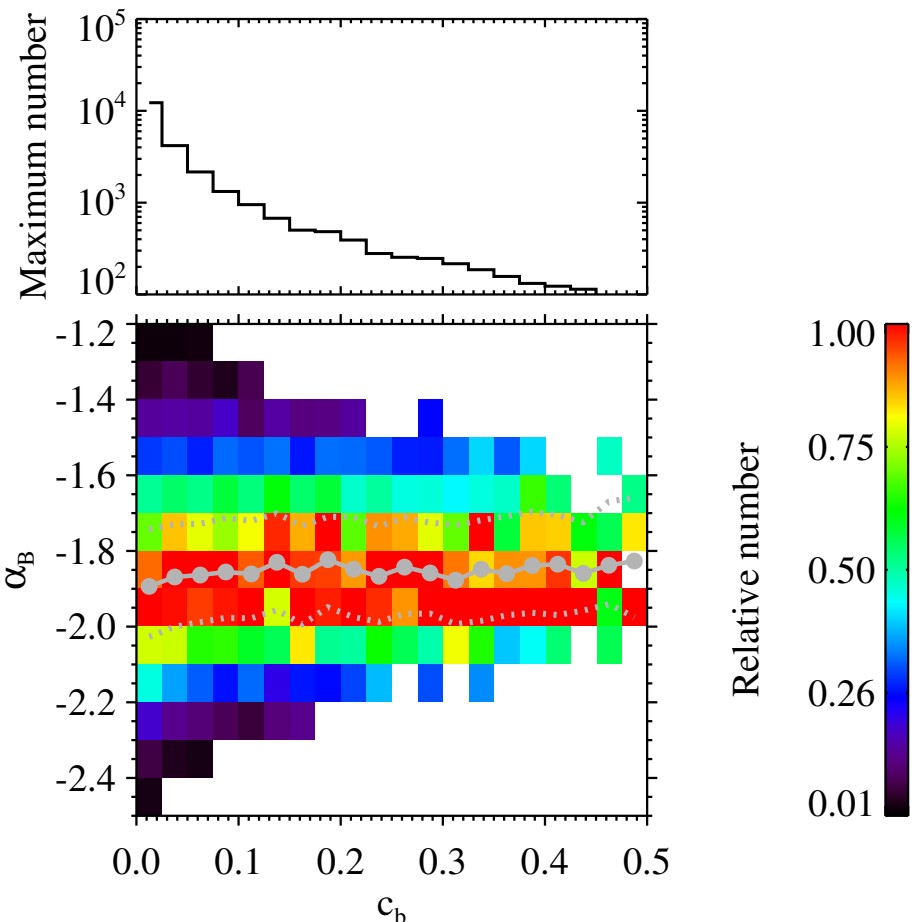

**Figure 7.** Joint distribution of magnetic compressibility $c_b$ and magnetic spectral index $\alpha_B$ for the selected 24,886 intermittent intervals in the same format as Figure 6.

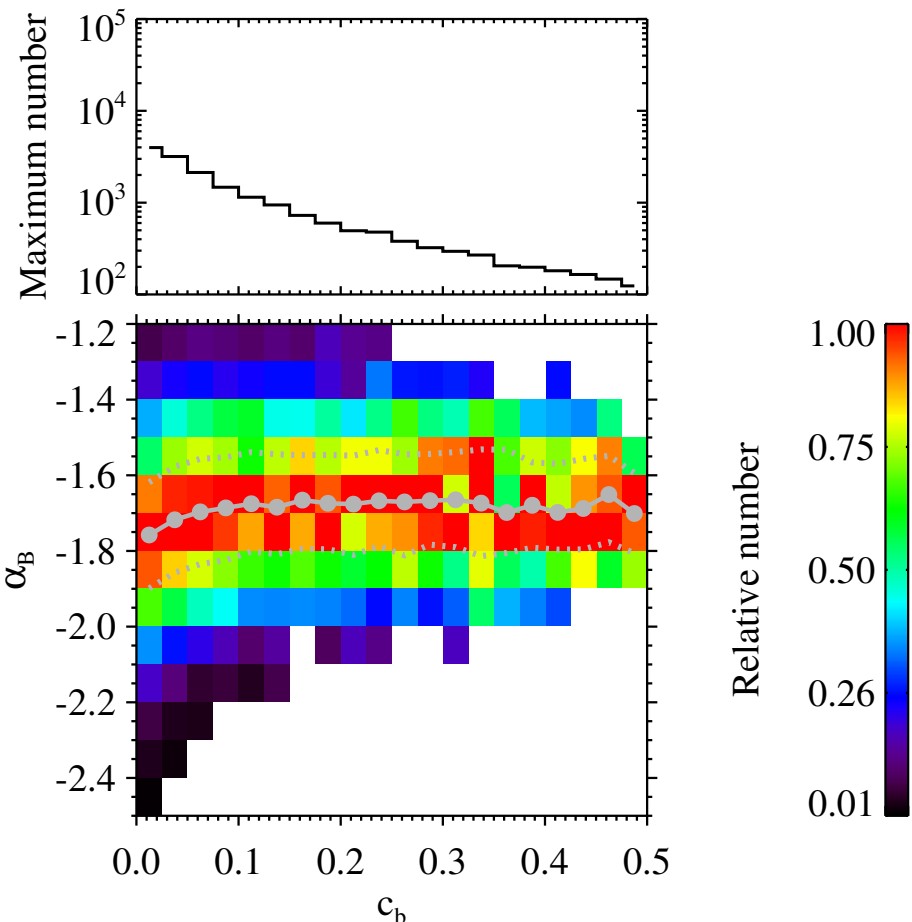

**Figure 8.** Joint distribution of magnetic compressibility $c_b$ and magnetic spectral index $\alpha_B$ for the selected 17,386 quiet intervals in the same format as Figure 6.

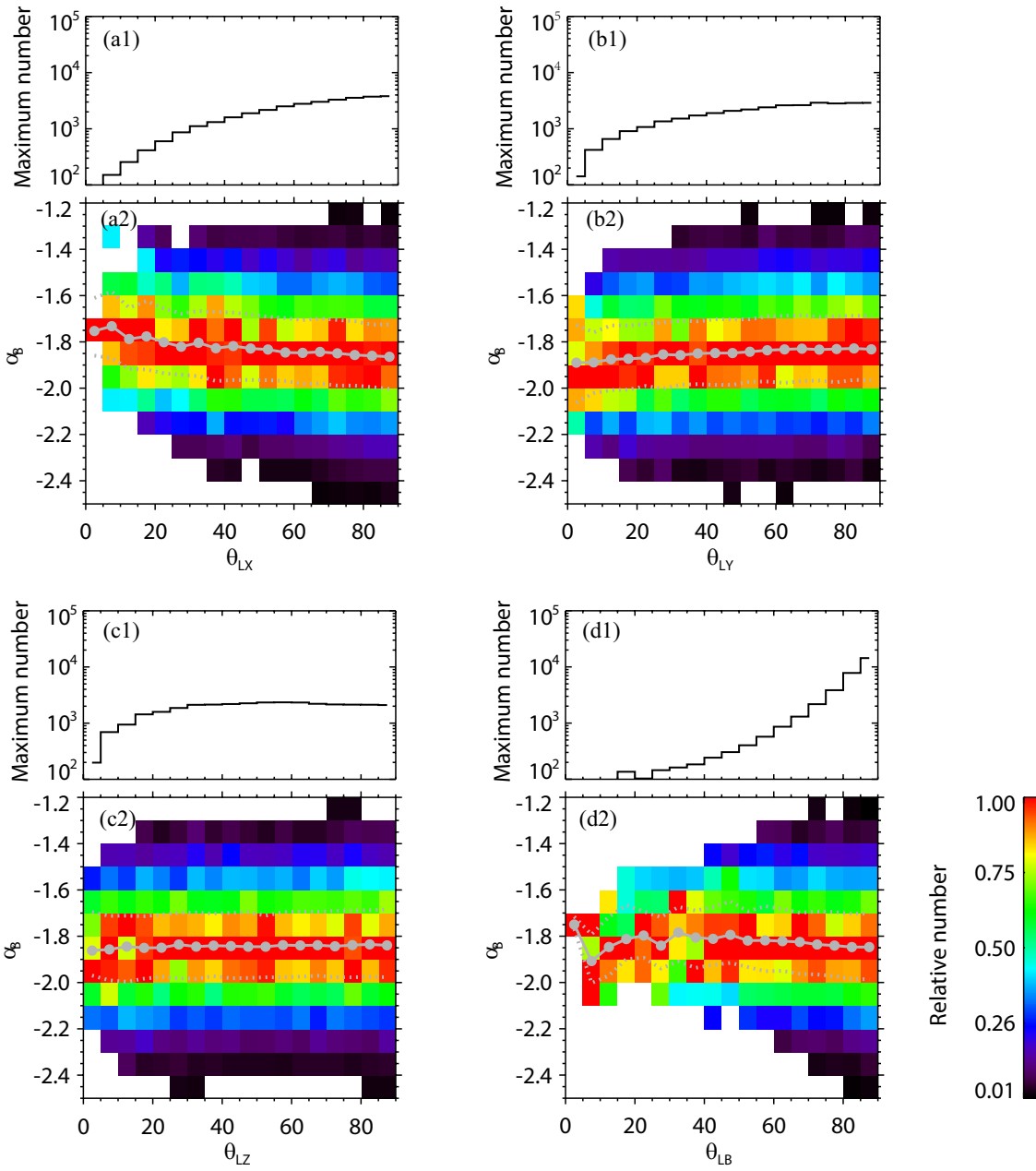

**Figure 9.** (a2) Joint distribution of $\theta_{LX}$ and magnetic spectral index $\alpha_B$ for the 33,261 intervals with $\lambda_1/\lambda_2 > 3$. For a given pixel, the color denotes relative number, which is the number of the cases normalized by the maximum number among the corresponding $\theta_{LX}$ bin. The maximum number of each bin is shown in panel (a1). The pixels containing no more than 10 cases are ignored. The gray solid circles represent average $\alpha_B$ in each $\theta_{LX}$ bin. The dotted gray lines represent the upper/lower quartiles. Panels (b1)(b2) are plotted in the same format as panels (a1)(a2) but for $\theta_{LY}$. Panels (c1)(c2) correspond to $\theta_{LZ}$. Panels (d1)(d2) correspond to $\theta_{LB}$.

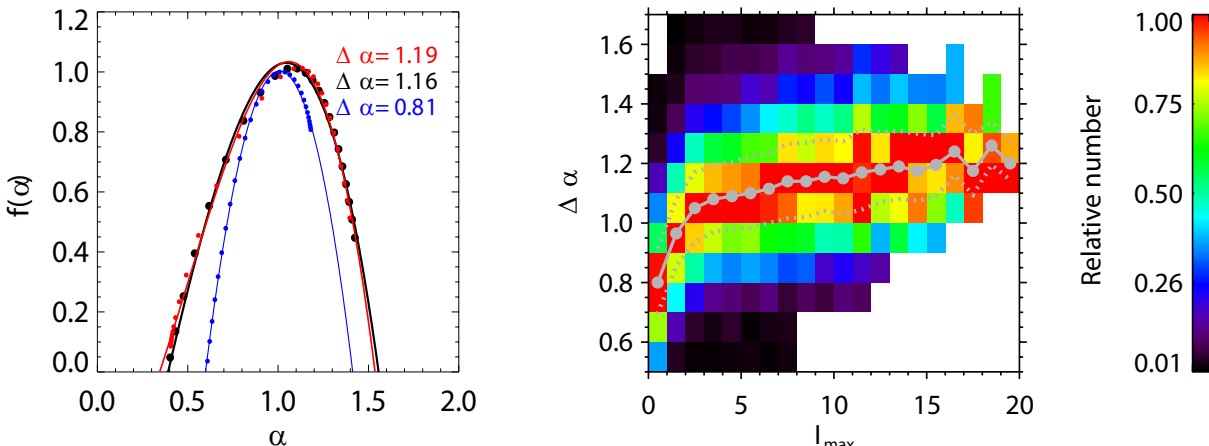

**Figure 10.** Left: Multifractal singularity spectra $f(\alpha)$ versus $\alpha$ observed in the slow wind (points) with red for the intermittent interval shown in Figure 4 ($I_{max} = 13.09$), black for the intermittent interval shown in Figure 2 ($I_{max} = 4.10$), and blue for the quiet interval shown in Figure 5 ($I_{max} = 1.44$). The solid lines denote the cubic polynomial fitting to the observations. The width of each singularity spectrum $\Delta\alpha = \alpha_{max} - \alpha_{min}$ is marked in the panel. Right: Joint distribution of $I_{max}$ and $\Delta\alpha$ for the 33,261 intervals with $\lambda_1/\lambda_2 > 3$. For a given pixel, the color denotes relative number, which is the number of the cases normalized by the maximum number among the corresponding $I_{max}$ bin. The pixels containing no more than 10 cases are ignored. The gray solid circles represent average $\Delta\alpha$ in each $I_{max}$ bin. The dotted gray lines represent the upper/lower quartiles.

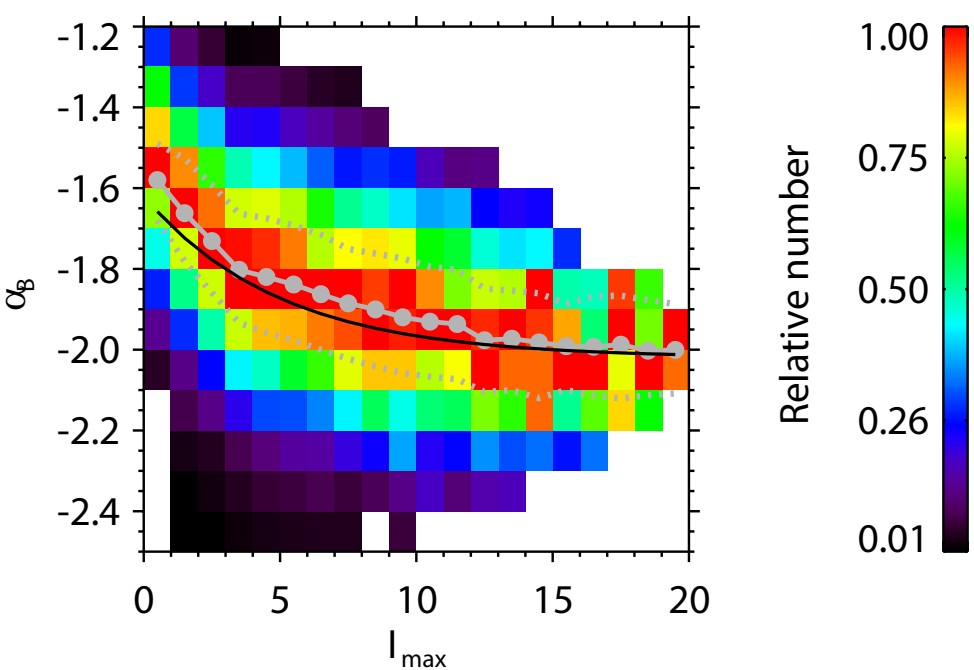

**Figure 11.** Joint distribution of intermittency level $I_{max}$ and magnetic spectral index $\alpha_B$ obtained from "linear detrending preparation" method plotted in the same format as the lower panel of Figure 6. The black curve is the exponential function $\alpha_B = 0.4\exp(-I_{max}/5) - 2.02$ adopted from Figure 6.

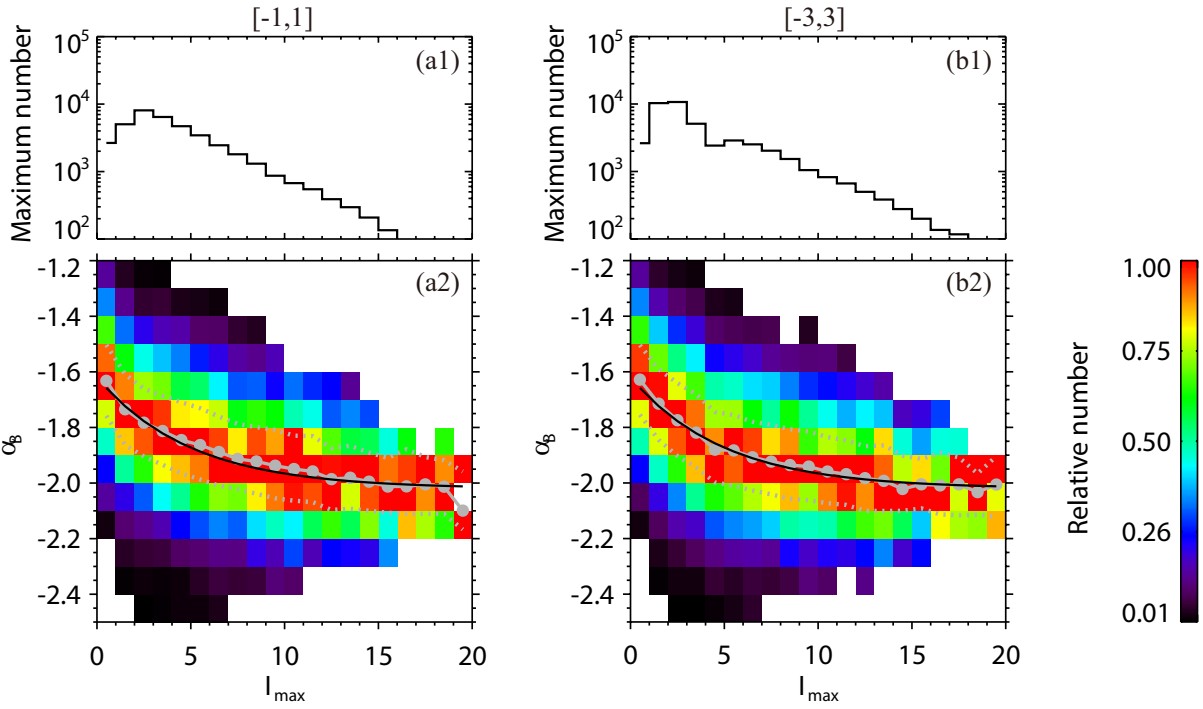

**Figure 12.** Panels (a1)(a2) and panels (b1)(b2) are plotted in the same format as Figure 6, but for the PVI thresholds $[-1, 1]$ and $[-3, 3]$ for identifying an intermittent interval, respectively. The black curves in panels (a2) and (b2) are both the exponential function $\alpha_B = 0.4 \exp(-I_{max}/5) - 2.02$, which is adopted from Figure 6.