# Peer review of "Effect of Intermittent Structures on the Spectral Index of Magnetic field in the Slow Solar Wind"

_Annales Geophysicae, 2022_

## Author Comment (AC1)

**Reply to "Comment from Anonymous Referee #1"**

We appreciate the referee's comments on our manuscript entitled "Effect of Intermittent Structures on the Spectral Index of Magnetic field in the Slow Solar Wind" [angeo-2022-28]. We have taken the referee's suggestions fully into account. In our response, each of the referee's suggestions is followed by the corresponding reply and revision marked in bold. Line numbers refer to the original submission. The comments have helped us to improve our manuscript and clarify the contents significantly. We are grateful for the referee's suggestions.

The manuscript "Effect of Intermittent Structures on the Spectral Index of Magnetic field in the Slow Solar Wind" by X. Wang and co-authors deals with the investigation of the intermittent properties of solar wind magnetic field fluctuations as measured by the spectral slope of their power spectral density. The paper is well written and the topic is within the scope of ANGEO. However, there are some missing aspects that need to be properly framed out and considered before it can be accepted for publication.

Major comments

1. The paper accounts for finding a relationship between the spectral exponent and the level of intermittency in slow solar wind streams observed at 1 AU by WIND. The results shown in the paper are not new since a close correspondence between intermittency and changes in the 2nd-order scaling properties has been well established. The main novelty is only the observed analytical relation (fit). I would suggest the authors to carefully revise the manuscript to clearly state this. There is a huge literature on the correction of the scaling properties due to intermittency as well as many improved cascade models have been proposed to revise the original Kolmogorov results.

**Reply: Thanks for the suggestion! We agree that the main novelty of this work is the observed analytical relation (fit). According to the suggestion, we revise the abstract and the main text to clearly state this point as follows.**

**In "Abstract", we revise two sentences as "... However, an analytical relationship between intermittency level and the magnetic spectral index has not been shown yet. ... Accordingly, an empirical relation is established between $\alpha_B$ and $I_{max}$ for the first time as $\alpha_B = 0.4 \exp(-I_{max}/5) - 2.02$. ..."**

**In the main text, we add the following sentences at Line 59: "From the previous studies mentioned above, people have realized that ... a close correspondence between intermittency and changes in the 2nd-order scaling properties has been well established. There is a huge literature on the correction of the scaling properties due to intermittency, and many improved cascade models have been proposed to revise the original Kolmogorov results. However, no analytical relationship between the magnetic spectral index and the level of intermittency**

has been shown so far. The main novelty of this work is that we show for the first time the analytical relationship between the magnetic spectral index and the level of intermittency by performing a fit on the observational results."

In "Conclusions", we replace the first sentence at Line 294 by the new one: "In this paper, we present for the first time the analytical relation between the magnetic spectral index $\alpha_B$ in the inertial range and the level of intermittency $I_{max}$ at the time scale of $\tau = 24$ s in the slow solar wind."

2. The authors claim, indeed, that the steeping of the spectrum is closely connected with intermittency. However, this could be only partially true since different spectral slopes are observed if looking along different directions with respect to the mean field. As the authors say there is a huge literature on the anisotropy of spectral slopes but they introduce a measure of the level of intermittency based on the trace of the PVI (so, something isotropic) and then also evaluate spectral slopes for the trace of the magnetic field fluctuations. Thus, my question is: how the presented results could be biased by anisotropy of magnetic field fluctuations? A possible check could be performed by looking at the dependence of the spectral slope across different directions as a function of the threshold crossing of the PVI along the different directions again. Would the results be robust or is there any dependence on the predominance of fluctuations along a specific direction?

In other words, what is the difference between an interval with $|PVI_j| > 2$ but $|PVI_k| < 2$ and an interval with $|PVI_j| > 2$ for all $j = x, y, z$?

Reply: Thanks for the suggestion! We agree that it is necessary to check whether the presented results will be biased by the anisotropy of magnetic field fluctuations or not. According to the suggestion, we add a subsection "4.2 Influence of anisotropy of magnetic field fluctuations" in "4. Discussion". In this subsection, we perform a check to see if the spectral index is dependent on the predominance of fluctuations along a specific direction. Here the direction of the predominant fluctuations is indicated by maximum variance (L) direction, which is obtained from the Minimum Variance Analysis. We show in Figure 9 the variations of the magnetic spectral index as a function of the angle between L and $i$ direction ($\theta_{Li}$) (where $i$ denotes the $x$-axis, $y$-axis, and $z$-axis of geocentric solar ecliptic coordinates), along with the variations of the spectral index versus the angle between L and the mean magnetic field direction of each interval ($\theta_{LB}$). It is found that the influence of the anisotropy of predominant fluctuations on the spectral index is not as significant as the influence of the level of intermittency ($I_{max}$) on the spectral index. Therefore, our results are robust. In subsection 4.2 of the new version of the manuscript, we describe this point in detail:

4.2 Influence of anisotropy of magnetic field fluctuations

[revised manuscript text omitted]

4. Another crucial point is the definition of the threshold above which an interval is considered intermittent, i.e., the PVI threshold. Indeed, the authors used a threshold of 2 since "The Gaussian distributions are located between the PVI range $[-2, 2]$". However, the definition of PVI is indeed a measure of the level of fluctuations with respect to an average level, i.e., something that resembles a standardization procedure. Did the authors performed the sensitivity of the results based on the choice of the threshold for identifying an intermittent interval?

**Reply: Thanks for the suggestion! According to the suggestion, we check the sensitivity of the results on the choice of PVI threshold. We add the following paragraph and a new figure in original line 316: "We also check the sensitivity of the results based on the choice of the threshold for identifying an intermittent interval. The threshold is changed from the original PVI range $[-2, 2]$ into two new ranges $[-1, 1]$ and $[-3, 3]$. The results are shown in Figure 12. The left panels and right panels correspond to the thresholds $[-1, 1]$ and $[-3, 3]$ for identifying an intermittent interval, respectively. They are plotted in the same format as**

**Figure 6.** The black curves in the lower two panels are both the exponential function $\alpha_B = 0.4\exp(-I_{max}/5) - 2.02$, which is adopted from **Figure 6.** It is found that the black curve obtained from the original threshold $[-2, 2]$ can still match the new results well. Therefore, our result shown in **Figure 6** is robust, and is not sensitive to the choice of the threshold for identifying intermittent intervals."

[Figure]

Figure 12: **Panels (a1)(a2) and panels (b1)(b2) are plotted in the same format as Figure 6, but for the PVI thresholds $[-1, 1]$ and $[-3, 3]$ for identifying an intermittent interval, respectively. The black curves in panels (a2) and (b2) are both the exponential function $\alpha_B = 0.4\exp(-I_{max}/5) - 2.02$, which is adopted from Figure 6.**

Additional detailed comments

Line 3: I would suggest to clarify that "an analytical/functional relationship..." has not been shown yet.

**Reply: Thanks! Revised.**

Line 4: the term "intermittency magnitude" could be biased by the definition, it would

be better to use the classical notation of "intermittency level".

**Reply: Thanks! We replace the term "intermittency magnitude" by "intermittency level" throughout the text.**

Line 59: again here an analytical relation has not been shown yet, while several cascade modes have found intermittency corrections to the spectral slope.

**Reply: Thanks for the suggestion! The original sentence is replaced by the new ones: "There is a huge literature on the correction of the scaling properties due to intermittency, and many improved cascade models have been proposed to revise the original Kolmogorov results. However, no analytical relationship between the magnetic spectral index and the level of intermittency has been shown so far."**

Lines 74-75: I am not sure the whole interval from 2005 and 2013 is characterized by an undisturbed solar wind (there are different transients indeed). I would suggest to state that the selected intervals all correspond to undisturbed solar wind conditions (if this is the case).

**Reply: Thanks! We delete the phrase "in the undisturbed solar wind", and the sentence is revised as "During this period, the WIND spacecraft was located at the Lagrangian point L1."**

Line 77: please change "∼" with "—".

**Reply: Thanks! Revised.**

Line 124: is it 15 s or 150 s as stated in line 114?

**Reply: The width of the intermittent structure is recorded as 15 s. We revise the sentence to clarify this point: "The two vertical dotted lines mark the beginning time ($t_B$=01:44:19) and ending time ($t_E$=01:44:34) of the intermittent structure, respectively. ... Accordingly, the width of this intermittent structure obtained from $t_E - t_B$, during which the condition $|PVI_z| > 2$ satisfies, is recorded as 15 s (5 data points)."**

Line 209: did you check that this is not an Alfvénic stream and then the spectrum should be f-3/2? If this is the case, this means that there is an intermittency correction.

**Reply: According to the suggestion, we check the Alfvénicity of this case,**

and find that it is not an Alfvénic interval with low normalized cross helicity $\sigma_c = 0.34$ and low Alfvén ratio $\gamma_A = 0.47$. We add the following sentence to clarify this point: "We check the Alfvénicity of this case, and find that it is not an Alfvénic interval with low normalized cross helicity $\sigma_c = 0.34$ and low Alfvén ratio $\gamma_A = 0.47$. It's worth noting that for an Alfvénic interval, if the magnetic spectrum scales as $f^{-5/3}$, an intermittency correction could be considered.

Line 232: which kind of discontinuities? This is important to understand which situation is presented.

Reply: The techniques used for the data analysis have strong influence on the classification of different types of discontinuities in the solar wind. We add the following sentences in Line 232: "In previous studies, the discontinuities in the solar wind have been identified mainly as rotational discontinuities (e.g., Neugebauer et al., 1984; Tsurutani and Ho, 1999; Wang et al., 2013; Liu et al., 2021). However, in some other studies, the discontinuities have been identified mainly as tangential discontinuities, depending on the different techniques used for data analysis (e.g., Horbury et al., 2001; Knetter et al., 2004; Riazantseva et al., 2005)."

Line 235: could the maximum value be biased by the anisotropy of fluctuations? I mean is this really representative of something new or simply a reflection of a spectrum of anisotropic fluctuations?

Reply: Thanks for the suggestion! According to the suggestion, we perform a check about whether the intermittency level $I_{max}$ could be biased by the anisotropy of fluctuations. In Figure R1 shown below, we present the joint distribution of $\theta_{LX}$ ($\theta_{LY}$, $\theta_{LZ}$, and $\theta_{LB}$ defined in the reply to "Major comments #2") and intermittency level $I_{max}$ in the similar format as Figure 9 shown above. From the average value of $I_{max}$ in each angle bin (gray dots) of panels (a)(b)(c) of Figure R1, we can see that the intermittency level $I_{max}$ appears to be not dependent on the direction of the predominant fluctuations. In panel (d), we see that the average $I_{max}$ ($\sim 4.0$) for $\theta_{LB} > 70°$ is sightly larger than the average $I_{max}$ ($\sim 2.6$) for $20° < \theta_{LB} < 60°$. However, from Figure 9(d2), it has been found that the spectral index $\alpha_B$ nearly does not change with $\theta_{LB}$. Accordingly, we add the following sentence in original line 235: "We have checked about whether the intermittency level $I_{max}$ could be biased by the anisotropy of fluctuations. It is found that the intermittency level $I_{max}$ appears to be not dependent on the direction of the predominant fluctuations (figure not shown here, since it is similar as Figure 9)."

[Figure]

Figure R1: **(a) Joint distribution of $\theta_{LX}$ and intermittency level $I_{max}$ for the 33,261 intervals with $\lambda_1/\lambda_2 > 3$. For a given pixel, the color denotes relative number, which is the number of the cases normalized by the maximum number among the corresponding $\theta_{LX}$ bin. The maximum number of each bin is shown in panel (a1) of Figure 9. The pixels containing no more than 10 cases are ignored. The gray solid circles represent average $I_{max}$ in each $\theta_{LX}$ bin. The dotted gray lines represent the upper/lower quartiles. Panels (b) is plotted in the same format as panel (a) but for $\theta_{LY}$. Panels (c) corresponds to $\theta_{LZ}$. Panel (d) corresponds to $\theta_{LB}$.**

Line 244: there is a recent literature on the scaling properties and intermittency levels with Parker Solar Probe (see papers by Alberti, Cuesta, Matthaeus).

**Reply: Thanks for the suggestion! We add the reference papers into the manuscript at line 244 as following: "Recently, there are several papers on the scaling properties and intermittency levels with Parker Solar Probe (e.g., Alberti**

et al., 2020; Cuesta et al., 2022; Sioulas et al., 2022).''

**References**

[revised manuscript text omitted]

---

## Author Comment (AC2)

**Reply to "Comment from Dr. Joseph Borovsky"**

We appreciate Dr. Joseph Borovsky for the comments on our manuscript entitled "Effect of Intermittent Structures on the Spectral Index of Magnetic field in the Slow Solar Wind" [angeo-2022-28]. We have taken the suggestions fully into account. In our response, each of the suggestions is followed by the corresponding reply and revision marked in bold. Line numbers refer to the original submission. The comments have helped us to improve our manuscript and clarify the contents significantly. We are grateful for the referee's suggestions.

This is a very interesting study, but this reader was at times confused about the methodology used in the data analysis. I am asking for a revised manuscript clarifying some of the data-analysis methods.

1. Throughout the paper, please make clear that the PSD is the magnetic PSD and that the spectral index is the magnetic spectral index.

**Reply: Thanks! Revised throughout the paper.**

2. It seems that the plasma data is only used to get the number density in order to put the magnetic data into Alfvén units. Can you clarify in the manuscript if that is true.

**Reply: According to the suggestion, we add the following sentence in original line 76 to clarify this point: "The plasma data is used here to get the bulk velocity for data selection and to get the proton number density in order to put the magnetic data into Alfvén units. In addition, the plasma data is also used to calculate Alfvénicity for the purpose of revealing the nature of intermittent structures."**

3. There is no description in the manuscript of how the time-series data was prepared prior to performong the FFT. Was the data windowed? Was the data interval de-trended? If not, then there is an extra discontinuity in the data that adds Fourier power to the PSD. Please add a description of the time-series preparation to the manuscript.

**Reply: Thanks for the suggestion! When performing the FFT on the components of magnetic field data, we use a simple rectangle window. We add a description of the time-series preparation prior to FFT in line 143 and in the caption of Figure 2, and also add a subsection and a new figure to compare the magnetic spectral index obtained from "no data preprocessing" method and "linear detrending preparation" method.**

**– Line 143: ".. the time series ... is Fourier transformed using the FFT method with a simple rectangle window. This method could introduce an extra**

discontinuity in the data that will add Fourier power to the magnetic PSD as mentioned by Borovsky (2012) and Borovsky and Burkholder (2020). In subsection 4.4, we apply a linear detrend to the data prior to Fourier transforming following Borovsky (2012), and make a comparison between the two methods."

– Caption of Figure 2: "The gray curve corresponds to the magnetic power spectrum by performing FFT on .. magnetic field data ... with a simple rectangle window."

– 4.4 Linear detrending to data prior to FFT

When performing the FFT on the components of magnetic field data, we use a simple rectangle window (hereinafter referred to as "no data preprocessing" method). This method could introduce an extra discontinuity in the data that will add Fourier power to the magnetic PSD as mentioned by Borovsky (2012) and Borovsky and Burkholder (2020). Following Borovsky (2012), we try applying a linear detrend to each data interval prior to Fourier transforming (hereinafter referred to as "linear detrending preparation" method), and compare the result with that in Figure 6 obtained from "no data preprocessing" method.

Figure 11 presents the joint distribution of intermittency level $I_{max}$ and magnetic spectral index $\alpha_B$ obtained from "linear detrending preparation" method plotted in the same format as the lower panel of Figure 6. The analytical relationship $\alpha_B = 0.4 \exp(-I_{max}/5) - 2.02$ adopted from Figure 6 is superposed on the figure as black curve for easier comparison. It is clear that when $I_{max} > 12$, the black curve coincides with the averaged magnetic spectral indices $\alpha_B$ (gray dots) well. However, when $I_{max} < 12$, the averaged magnetic spectral indices $\alpha_B$ (gray dots) obtained from "linear detrending preparation" method appear to be larger than that obtained from "no data preprocessing" method denoted by the black curve. The differences between them are about $0.01 - 0.06$. This is consistent with Borovsky (2012), which mentioned that the "no data preprocessing" method leads to spectral indices slightly steeper. When looking at the upper/lower quartiles, we notice that the distribution of $\alpha_B$ in a $I_{max}$ bin obtained from "linear detrending preparation" method (e.g., $\alpha_B = -1.90^{+0.15}_{-0.14}$ at $I_{max} = 8.5$) is slightly wider than that obtained from "no data preprocessing" method (e.g., $\alpha_B = -1.93^{+0.13}_{-0.12}$ at $I_{max} = 8.5$). The wider distribution for the "linear detrending preparation" method is also consistent with Borovsky (2012). Accordingly, we suggest that when using different data preprocessing methods, the magnetic spectral index slightly changes, but our results about the trend of the magnetic spectral index $\alpha_B$ versus the intermittency level $I_{max}$ and the contribution of the intermittency on the magnetic spectra are robust.

[Figure]

Figure 11: **Joint distribution of intermittency level $I_{max}$ and magnetic spectral index $\alpha_B$ obtained from "linear detrending preparation" method plotted in the same format as the lower panel of Figure 6. The black curve is the exponential function $\alpha_B = 0.4\exp(-I_{max}/5) - 2.02$ adopted from Figure 6.**

4. The PSDs in the figures are in units of velocity, meaning that the time series of the magnetic field in Alfven units was used in the FFT. The magnetic-field data has a resolution of 1/11 sec while the plasma data has a resolution of 3 sec. How were the values of the number density chosen to put the magnetic-field data into Alfvén units. One density value for the entire time series interval? Changing the density value every 3 seconds in the time series?

**Reply: We add the following sentences to clarify about how are the values of number density chosen to put the magnetic-field data into Alfvén units.**

– **Original line 118: "The magnetic field data are transformed into Alfvén units (i.e., $\mathbf{B}/\sqrt{\mu_0 m_p \langle n_p \rangle}$ with $\mu_0$ being susceptibility, $m_p$ being proton mass, and $\langle n_p \rangle$ being the average proton number density of the $\sim$5-min interval)."**

– **Original line 143: "The high-resolution magnetic field data are first transformed into Alfvén units (i.e., $\mathbf{B}/\sqrt{\mu_0 m_p \langle n_p \rangle}$ with $\langle n_p \rangle$ being the average proton number density of each interval). ... the time series of each component of the**

high-resolution magnetic field data in Alfvén units is Fourier transformed ..."

— Caption of Figure 2: "The magnetic field is plotted in Alfvén units (i.e., $B/\sqrt{\mu_0 m_p \langle n_p \rangle}$ with $\mu_0$ being susceptibility, $m_p$ being proton mass, and $\langle n_p \rangle$ being the average proton number density of this interval). ... The gray curve corresponds to the magnetic power spectrum by performing FFT on the 1/11-s-resolution magnetic field data in Alfvén units obtained still from $B/\sqrt{\mu_0 m_p \langle n_p \rangle}$."

5. When putting the magnetic field into Alfvén units, if one value of number density for the entire interval is not chosen, how different is the spectral index of the magnetic field in Alfvén units versus the spectral index of the magnetic field in nT? I would worry that noise in the density measurements (particularly in the WIND 3-sec onboard moments) would spoil the spectral-index value. Can you comment on this possibility in the manuscript.

**Reply: We use the ensemble average of proton number density for each selected interval when putting the magnetic field into Alfvén units. We emphasis that we use the one value of number density in the text and add the following sentences in original line 143: "When putting the magnetic field into Alfvén units, we use one value of proton number density, which corresponds to the ensemble average of proton number density $\langle n_p \rangle$ for each selected interval. By doing so, we avoid the contamination of the noise in density measurements on the magnetic spectral-index value, which would be resulted from using the density value changing every 3 seconds."**

6. $> 42,000$ intervals were examined but only 24,886 intervals were used for the statistics. That means almost half of the intervals were rejected. Besides having a higher fitting error, were there any trends to what was rejected and what was accepted?

**Reply: Besides having a higher fitting error, we also eliminated the cases during which the energy of the fluctuations is not dominated by the intermittent structure imbedded in the center of it as mentioned in original line 136. We add the following sentences in original line 296 to clarify this point: "We examine 56,398 intermittent structures preliminarily by using the criterion $|PVI_i| > 2$ ($i = x, y,$ or $z$), with $t_B$ and $t_E$ being the beginning and ending instants of a structure, respectively. However, for more than half of them, the maximum $I$ ($I_{max}$) during $[t_B, t_E]$ (as marked by the two vertical dotted lines in Figure 2) is not the maximum $I$ during the corresponding plotted interval $[t_B - 150s, t_E + 150s]$ (as the whole plotted interval in Figure 2). It means that outside $[t_B, t_E]$, there exist some other structures with even higher level of intermittency during the interval $[t_B - 150s, t_E + 150s]$. We eliminate this kind of intervals, during which the energy of the fluctuations is not dominated by the intermittent structure imbedded in**

the center of it. In this way, we avoid the duplicate selection of the cases, and also guarantee that both the intermittency level $I_{max}$ and the magnetic spectral index $\alpha_B$ are closely related to the intermittent structure imbedded in the middle of each interval. Then we obtain 25,912 intermittent intervals. Subsequently, the cases with higher fitting error of the magnetic power spectra ($\Delta_{\alpha_B}/\alpha_B > 5\%$) are eliminated, and 24,886 intermittent intervals are reserved for the statistical analysis."

7. When the "width" of an intermittent spot is measured (line 264 and Figure 3), what are the units? Data points at 1/11-sec resolution? Data points at 3-sec resolution? Please clarify for the reader.

Reply: Thanks for the suggestion! We clarify this point both in line 164 and in the caption of Figure 3 as following: "... present the joint distribution of their width in units of data points and ... . Here, the width in units of data points for an intermittent structure is obtained from $t_E - t_B$, during which the condition $|PVI_i| > 2$ satisfies ($i = x, y,$ or $z$), divided by the time resolution $\Delta t = 3$ s."

**References**

Borovsky, J. E.: The velocity and magnetic field fluctuations of the solar wind at 1 AU: Statistical analysis of Fourier spectra and correlations with plasma properties, Journal of Geophysical Research (Space Physics), 117, A05104, https://doi.org/10.1029/2011JA017499, 2012.

Borovsky, J. E. and Burkholder, B. L.: On the Fourier Contribution of Strong Current Sheets to the High-Frequency Magnetic Power SpectralDensity of the Solar Wind, Journal of Geophysical Research (Space Physics), 125, e27307, https://doi.org/10.1029/2019JA027307, 2020.